# Advances in Medical Image Segmentation: A Comprehensive Review of Traditional, Deep Learning and Hybrid Approaches

**DOI:** 10.3390/bioengineering11101034

**Published:** 2024-10-16

**Authors:** Yan Xu, Rixiang Quan, Weiting Xu, Yi Huang, Xiaolong Chen, Fengyuan Liu

**Affiliations:** 1School of Electrical, Electronic and Mechanical Engineering, University of Bristol, Bristol BS8 1QU, UK; yan.xu@bristol.ac.uk (Y.X.); rixiang.quan@bristol.ac.uk (R.Q.); w.xu.2021@bristol.ac.uk (W.X.); 2Bristol Medical School, University of Bristol, Bristol BS8 1UD, UK; yfyi.huang@bristol.ac.uk; 3Department of Mechanical, Materials and Manufacturing Engineering, University of Nottingham, Nottingham NG7 2RD, UK; xiaolong.chen@nottingham.ac.uk

**Keywords:** image segmentation, medical image processing, deep learning, biomedical engineering, medical imaging, diagnostic imaging

## Abstract

Medical image segmentation plays a critical role in accurate diagnosis and treatment planning, enabling precise analysis across a wide range of clinical tasks. This review begins by offering a comprehensive overview of traditional segmentation techniques, including thresholding, edge-based methods, region-based approaches, clustering, and graph-based segmentation. While these methods are computationally efficient and interpretable, they often face significant challenges when applied to complex, noisy, or variable medical images. The central focus of this review is the transformative impact of deep learning on medical image segmentation. We delve into prominent deep learning architectures such as Convolutional Neural Networks (CNNs), Fully Convolutional Networks (FCNs), U-Net, Recurrent Neural Networks (RNNs), Adversarial Networks (GANs), and Autoencoders (AEs). Each architecture is analyzed in terms of its structural foundation and specific application to medical image segmentation, illustrating how these models have enhanced segmentation accuracy across various clinical contexts. Finally, the review examines the integration of deep learning with traditional segmentation methods, addressing the limitations of both approaches. These hybrid strategies offer improved segmentation performance, particularly in challenging scenarios involving weak edges, noise, or inconsistent intensities. By synthesizing recent advancements, this review provides a detailed resource for researchers and practitioners, offering valuable insights into the current landscape and future directions of medical image segmentation.

## 1. Introduction

Medical image segmentation is a crucial process in medical imaging, forming the foundation for accurate diagnosis, treatment planning, and quantitative analysis across various clinical applications [1]. It involves partitioning an image into distinct regions that correspond to anatomical structures or pathological areas, such as organs, tissues, or lesions [2]. This segmentation allows for precise interpretation of medical images, supporting critical tasks such as tumor localization, organ boundary delineation, and pre-surgical planning [3]. Given its significance, the accuracy of segmentation has a direct impact on clinical outcomes, making it an indispensable element in the workflow of modern healthcare systems.

For decades, traditional segmentation methods (such as thresholding [4], edge-based techniques [5], region-based approaches [6], clustering-based methods [7], and graph-based segmentation [8]) have been widely employed. These methods rely on predefined rules and intensity-based operations, making them relatively efficient and interpretable. However, their performance often degrades when confronted with complex medical images containing noise [9], intensity variations [10], or ambiguous boundaries [11]. These limitations underscore the need for more adaptive and robust approaches to segmentation.

In recent years, deep learning has revolutionized the field of medical image segmentation by enabling automatic feature extraction and handling of complex, high-dimensional data [12]. Architectures such as Convolutional Neural Networks (CNNs) [13,14], Fully Convolutional Networks (FCNs) [15], U-Net [16,17], and Recurrent Neural Networks (RNNs) [18] have become central to this transformation, delivering superior performance in pixel-wise segmentation tasks. Generative Adversarial Networks (GANs) [19] and Autoencoders (AEs) [20] have further extended the boundaries of segmentation, enhancing both accuracy and adaptability, particularly in scenarios where traditional methods falter due to noise or high variability in the images.

Despite these advances, deep learning also presents challenges, such as the demand for large, annotated datasets and significant computational resources. To address these limitations, hybrid approaches [21] combining traditional methods with deep learning have gained momentum. These models harness the strengths of both paradigms, improving segmentation performance in challenging medical images by blending rule-based techniques with adaptive feature learning.

This review aims to offer a comprehensive exploration of medical image segmentation techniques. It begins by critically evaluating traditional medical image segmentation methods. It then delves into deep learning approaches, focusing on key architectures such as CNNs, FCNs, U-Net, GANs, RNNs, and AEs and their specific contributions to medical image segmentation. Finally, this review discusses the applications of hybrid approaches that integrate deep learning with traditional segmentation methods in medical image segmentation. By synthesizing recent developments, the review provides practical insights into the strengths and limitations of these combined techniques, offering a valuable reference for ongoing research and clinical practice.

## 2. Traditional Segmentation Techniques

For a long time, traditional segmentation techniques have served as the foundation of medical image analysis, offering a range of methods that vary in complexity and applicability. These techniques (including thresholding, edge detection, region-based segmentation, clustering, and graph-based methods) have proven effective in many scenarios. However, they often face limitations when dealing with the inherent complexity of medical images. Figure 1 provides an overview of the traditional segmentation techniques discussed in this session.

### 2.1. Thresholding Techniques

Thresholding stands as a pivotal technique within the realm of image segmentation methodologies. Its primary objective is to transmute a grayscale image into its binary counterpart, where each pixel adopts one of two distinct values, typically represented as 0 or 1, which can be visually interpreted as black or white, respectively. Thresholding can be more intuitively described as follows:(1)$$B\left(x,y\right)=\left\{\;\begin{array}{c} 1,\;\;if\;I\left(x,y\right)\geq T\\ 0,\;\;if\;I\left(x,y\right)<T \end{array}\right.$$
where

*I*(*x*,*y*) is the pixel value of the grayscale image at position (*x*,*y*);*B*(*x*,*y*) is the pixel value of the binary image at position (*x*,*y*);*T* is the chosen threshold.

Within image processing, thresholding methodologies can be taxonomically bifurcated into global and local thresholding, contingent upon the modality of threshold selection and its subsequent application [22].

#### 2.1.1. Global Thresholding Techniques

Global thresholding hinges on adopting a singular, invariant threshold, meticulously ascertained from the homogeneous attributes pervasive throughout the image or inferred from the comprehensive histogram. Typically, the global thresholding technique is employed when there is a distinct difference in the grey scale distribution between the foreground (or region of interest) and the background [22]. Notable methods of global thresholding encompass Ostu’s Method, Iterative Thresholding, Minimum Error Thresholding, and Entropy-based Thresholding.

Ostu’s method: The Otsu thresholding technique determines the optimal threshold by maximizing the variance between the grey levels of an object and its background [23]. Initially, it requires converting the image into greyscale, which consists of 256 grey levels ranging from 0 (black) to 255 (white). The method aims to identify a grey value threshold that separates the background (higher values) from the foreground (lower values). The ideal threshold is where this variance is at its maximum, enhancing the distinction between black and white in the binarized image.

Iterative thresholding: Iterative thresholding is a technique for binarizing images by repeatedly adjusting the threshold based on the mean grey scale values of the foreground and background [24]. Starting with an initial threshold *T_1_* (typically the average grey value of the image), the image is divided into two groups: pixels above *T_1_* (background) and below *T_1_* (foreground). The means of these groups are calculated to derive a new threshold *T_2_*. This process is repeated, updating *T_1_* with *T_2_* until the change between thresholds is within a small tolerance *T_0_* (usually *T_0_* = 0.5 or 1), at which point *T_2_* is deemed optimal [25].

Minimum Error Thresholding: The Minimum Error Thresholding technique, also known as the Kittler–Illingworth method, operates on the premise that the pixel grey values in an image’s foreground and background normally distribute. This method aims to find a threshold T that minimizes the total classification error after segmentation. This error is measured using the probability density functions of each segment, factoring in their probabilistic magnitude and variance [26].

Entropy-based thresholding: Pun’s entropy thresholding method maximizes the a posteriori entropy of segmented objects and backgrounds in greyscale images to enhance information content [27].

#### 2.1.2. Local Thresholding Techniques

Uniform thresholding methodologies may fail to yield optimal results under conditions of heterogeneous illumination attributable to variables such as shadows or directional lighting. In contrast, local thresholding techniques demonstrate enhanced suitability for such scenarios, accommodating the non-uniform light distribution [28]. These local approaches necessitate the implementation of more sophisticated algorithms that meticulously account for the local attributes of each segment within the image. Specifically, local thresholding assigns values contingent upon the statistical parameters prevalent within each discrete neighborhood, considering elements such as luminance, contrast, and textural nuances. This means that each pixel point in an image is classified based on the characteristics of its surrounding pixels [22,29]. Among the array of local adaptive thresholding methodologies are Niblack’s Method, Sauvola’s Method, and Bernsen’s Method.

Niblack’s Method: Niblack’s technique is a notable method in local adaptive thresholding. It calculates the local mean (m) and standard deviation (s) of pixel values within a specific window centered on each pixel [30]. The method leverages the mean to assess local brightness and the standard deviation to gauge contrast or texture, dynamically setting thresholds for effective image segmentation [31].

Sauvola’s Method: Niblack’s method struggles with low-texture backgrounds where fine details may surpass the set threshold. Sauvola enhanced this approach to better handle varying backgrounds and lighting by incorporating the dynamic range of the standard deviation, R, into the threshold calculation. This modification allows the method to adaptively amplify the standard deviation’s dynamic range [31].

Bernsen’s Method: Bernsen’s method focuses on adaptive segmentation, leveraging the local contrast of pixel regions to differentiate between high-contrast areas—indicative of edges or text—and low-contrast regions, typically homogeneous backgrounds [22]. This approach excels at detecting subtle image variations and highlighting key features against diverse backgrounds [32].

### 2.2. Edge-Based Segmentation

An edge is a boundary between two homogeneous regions and is a local variation in the intensity of an image. Edge detection is the process of identifying and locating sharp discontinuities in an image. In edge-based segmentation methods, the contours of the objects in the image and the boundaries between the objects and the background are first detected. Then, the edges are joined to form object boundaries to segment the desired region [33]. Discontinuity-based segmentation methods can detect sudden changes in intensity values [34]. The essence of edge detection is to detect the locations where the image characteristics have changed.

In edge-based image segmentation techniques, the process typically unfolds in a sequence of well-defined steps. It begins with image pre-processing, which sets the stage for subsequent analysis. This is followed by edge detection, a crucial phase for identifying potential boundaries within the image. Next, non-maximum suppression is applied to refine these edge detections, emphasizing the most significant edges while diminishing weaker ones. Subsequently, thresholding is performed, converting the gradient image into a binary format to distinctly separate edges from non-edges. Immediately following this step is post-processing, which further refines the segmented edges, potentially filling gaps or smoothing irregularities. The culmination of this process is the final image segmentation, where the image is partitioned into segments based on the identified and processed edges [35].

Edge detection is crucial in edge-based segmentation. Image edges are characterized by two fundamental elements: direction and magnitude. Along the direction of the edge, pixel value transitions are typically gradual. In contrast, changes in pixel values perpendicular to the edge are more pronounced. Due to these characteristics, both first and second-order derivatives are commonly used to describe and detect edges in images.

First-order differential operators like the Roberts Cross Operator [36], Prewitt Operator [37], Sobel Operator [38], and Canny Edge Detector [39,40] detect edges by highlighting gradients in pixel intensities. Second-order differential operators, such as the Laplacian Operator [41] and Laplacian of Gaussian [42], are also frequently utilized. These operators work by convolving a specific template matrix with the pixel value matrix of the image, effectively calculating gradients or curvatures at each pixel [43].

Each edge detection operator has distinct advantages and limitations (Table 1), making them suitable for different application scenarios.

The Roberts Cross operator is known for its simplicity and speed due to its small kernel size, making it particularly well-suited for less noisy images. However, it is highly sensitive to noise and offers only moderate performance in precise edge localization.

Both the Prewitt and Sobel operators utilize pixel neighborhood differencing. The Prewitt operator is more sensitive to edge direction, while the Sobel operator offers improved noise suppression, although this comes at the cost of potentially blurring edges slightly. The Canny operator, recognized for its high precision, is a multi-stage edge detection process that effectively suppresses noise and accurately locates edges, but it requires more computational resources [48].

On the other hand, the Laplacian operator, which uses second-order derivatives, emphasizes image details. Although it is more prone to noise interference, it excels in enhancing finer details. The Laplacian of Gaussian operator combines Gaussian smoothing with the Laplacian operator to mitigate noise effects through pre-smoothing, making it suitable for scenarios requiring precise edge localization in noise-sensitive environments [35].

### 2.3. Region-Based Segmentation

In image processing and computer vision, region-based segmentation is a frequently utilized approach. It segments images into areas that exhibit similar properties, enabling a more nuanced analysis of the visual content [49].

Region-based segmentation in image processing and computer vision is fundamentally guided by two key principles: pixel similarity and spatial proximity. This technique hinges on the premise that pixels within a given region share common characteristics, such as color, brightness, or texture, ensuring consistency within the segment. Beyond mere attribute similarity, the spatial closeness of pixels is also a critical factor. Pixels that are adjacent or near each other are typically more likely to be classified within the same region, reinforcing the segmentation’s coherence [50]. Two principal methods dominate the field of region-based segmentation in segmenting images into distinct regions based on pixel similarity: Seeded Region Growing (SRG) and Split and Merge.

#### 2.3.1. Seeded Region Growing

SRG, which was proposed by Adams and Bischof, stands out for its robustness, speed, and lack of parameter tuning requirements [51]. These characteristics enable the development of highly effective algorithms, making SRG versatile and suitable for a diverse array of images [52].

The SRG technique in image analysis can be succinctly outlined in four key steps [53]:Seed Point Selection: Initiate by identifying one or more seed points within the image. These points are chosen based on specific attributes such as brightness, color, or texture, which serve as a foundation for region growth;Criteria Determination: Define the parameters that will guide the region’s growth. This could involve setting thresholds for color differences, brightness similarities, or texture consistencies between adjacent pixels;Regional Expansion: Evaluate the pixels surrounding each seed point. Based on the established criteria, decide whether to integrate these pixels into the current region. This inclusion leads to a gradual expansion of the region;Iterative Process: Continue this procedure iteratively, adding new pixels to regions as appropriate. The process concludes when no further pixels can be assimilated into any region.

#### 2.3.2. Split and Merge

The split-and-merge algorithm is a straightforward approach to region detection, operating through a hybrid methodology [54]. This process alternates between subdividing the initial image into a collection of disjoint regions and merging those regions. The subdivision can be achieved either by a top-down splitting procedure or a bottom-up merging procedure, with both operations guided by segmentation conditions defined through a Boolean predicate *P* [55]. The basic process of splitting and merging includes the following steps [55,56]:
5.Let R represent the whole image with different objects. Segmentation can be thought of as dividing R into n subregions $R_{1}$, $R_{2}$, …, $R_{n}$ as a process;6.Split any region $R_{n}$ into four almost equal regions where $P\left(R_{n}\right)=FALSE$;7.If $P\left(R_{1}\cup R_{2}\cup \ldots \cup R_{n}\right)=TRUE$, then consider any two or more neighboring subregions $R_{1}$, $R_{2}$, …, $R_{n}$, and then merge the n regions into a single region;8.Repeat steps 2–3 until no further splitting and merging is possible.


### 2.4. Clustering-Based Segmentation

Clustering-based techniques serve as invaluable tools for image segmentation, facilitating the categorization of pixels with akin characteristics into distinct clusters. A fundamental approach to data clustering involves the systematic division of data points into clusters predicated on predetermined similarity or dissimilarity metrics. Subsequently, each cluster is labeled, ensuring that elements within the same cluster exhibit greater similarity to each other in contrast to elements residing in different clusters [33,57].

To provide a comprehensive framework for clustering methodologies, Fraley and Raftery [58] have proposed a dichotomy, classifying clustering methods into two distinct groups: hierarchical and partitional techniques. Figure 2 visually offers a structured overview of the multifaceted landscape of clustering methodologies.

#### 2.4.1. Hierarchical Clustering

In the realm of hierarchical clustering methodologies, clusters emerge through an iterative process of pattern segregation, employing either a top-down or a bottom-up approach. Such algorithms tackle the clustering challenge by constructing a binary tree-like data structure known as a dendrogram (Figure 3) [59]. Hierarchical methods bifurcate into two primary categories: agglomerative hierarchical clustering and divisive hierarchical clustering [60].

Agglomerative hierarchical clustering commences at the most granular level, with each cluster encapsulating a singular data entity. This method progressively amalgamates pairs of clusters, thereby cultivating a hierarchical structure from the bottom up. Conversely, the divisive approach adopts a top-down strategy, systematically fragmenting a comprehensive cluster, encompassing all entities, into increasingly smaller subdivisions. This division persists until each entity is isolated into its own cluster or until pre-defined termination criteria are satisfied [57].

Hierarchical clustering methods employ a greedy approach where once a data item is allocated to a cluster, it is not reassessed, leading to an inability to correct misclassifications and reduced robustness, especially in the presence of noise and outliers. These methods do not optimize any objective function during clustering, which is problematic for overlapping clusters and compromises performance [7]. Additionally, hierarchical clustering often requires prior knowledge of the number of clusters, complicating cluster formation. Issues also arise with spherical cluster formation and potential distortion of the hierarchical structure [61]. Furthermore, the time complexity of hierarchical methods makes them computationally expensive, particularly for high-dimensional datasets like images [7].

#### 2.4.2. Partitional Clustering

Partitional clustering methodologies categorize data items into distinct clusters by optimizing an objective function to enhance the similarity within each cluster relative to others. This process measures the similarity of each data item across various clusters, aiming to minimize a within-cluster similarity criterion, often quantified using Euclidean distance. This objective function evaluates the efficacy of each cluster, seeking the most representative configuration. Like hierarchical clustering, partitional clustering requires specifying the number of clusters in advance. A key feature is the comprehensive allocation of every data item to a specific cluster, even if an item lies far from its cluster’s center, which can result in distorted shapes or inaccuracies, especially in noisy or outlier-heavy datasets. Partitional clustering includes soft clustering (exemplified by Fuzzy c-means clustering (FCM) [62,63]) and hard clustering (notably the k-means algorithm [64,65,66]).

The main advantage of partitioning algorithms is their ability to iteratively improve clustering quality, a feature not found in hierarchical clustering methods [59]. However, these algorithms have several limitations. They often lack adequate cluster descriptors, which depend on a predetermined number of clusters. They are also highly sensitive to initial conditions and can be significantly affected by noise and outliers. Additionally, partitioning algorithms often struggle with clusters that vary in size and density or have non-convex shapes [67].

### 2.5. Graphic-Based Segmentation

GTAs are pivotal in the field of medical image segmentation. Consider a graph $G=\left(V,E\right)$, where $V=\left\{v_{1},\ldots ,\;v_{n}\right\}$ represents a set of vertices corresponding to image elements, which may be pixels or regions within Euclidean space. The set E comprises edges connecting certain pairs of neighboring vertices. Each edge $(v_{i},\;v_{j})\;\epsilon \;E$ is assigned a weight $\omega (v_{i},\;v_{j})$ that quantifies a specific property between the two connected vertices [68]. In the context of image segmentation, an image is divided into mutually exclusive components, such that each component A forms a connected subgraph $G^{\prime }=(V^{\prime },\;E^{\prime })$, where $V^{\prime }\subseteq V$ and $E^{\prime }\subseteq E$, with $E^{\prime }$ containing only edges formed from nodes in $V^{\prime }$ [69]. The union of the vertices in all subgraphs equals the complete set of vertices in the original graph. Each subgraph comprises a collection of vertices that exhibit a strong affinity towards each other, indicative of their interconnectedness and similarity [70]. Figure 4 shows graphically the relationship between image segmentation and graph partitioning.

#### 2.5.1. Minimal Spanning Tree (MST) Based Methods

MST is a concept from graph theory where a subset of edges forms a tree that includes every vertex of the graph and minimizes the total weight of these edges. Specifically, for a graph G, the spanning tree T is defined as $T=\left(V,\;E^{\prime }\right)$, where $E^{\prime }\subseteq E$. This principle is highly relevant in image segmentation, where pixels or regions are treated as nodes, and edges represent some measure of the difference between these nodes.

The construction of MSTs for image segmentation generally utilizes two well-known algorithms: Kruskal’s and Prim’s. Kruskal’s algorithm [71] begins by sorting all the edges of the graph by increasing weight and incrementally adding them to a growing forest of MSTs, ensuring no cycles are formed. In contrast, Prim’s algorithm [72] initiates from a single node and incrementally adds the least costly adjacent edge to expand the tree, ensuring connectivity at minimal cost.

#### 2.5.2. Graph Cuts

Graph cuts treat the segmentation problem as a graph partitioning problem where an image is converted into a graph $G=\left(V,E\right)$ with nodes V representing pixels or voxels and edges E representing the neighborhood relationships between them.

In this graph-based segmentation framework, there are two distinct types of vertices [73]. The first type, referred to as neighborhood nodes, corresponds to individual pixels within the image. The second type comprises terminal nodes, specifically the source (s) and sink (t), forming what is commonly known as an s−t graph. In such a graph, the source nodes (s) typically represent the objects of interest in an image, while the sink nodes (t) denote the background. There are also two principal types of edges in this graph structure [74]. The first, known as n−links, connects neighboring pixels to each other within the image, facilitating the assessment of pixel similarity. The second type, t−links, connects the terminal nodes to these neighborhood nodes, linking each pixel to either the source or the sink based on predefined criteria. Table 2 provides a comprehensive summary of the various graphic cutting methods.

#### 2.5.3. Markov Random Fields (MRF)

The segmentation of an image can fundamentally be described as the task of assigning labels to pixels [82]. Within the probabilistic framework, this labeling process is modeled using a MRF, where the optimal labels are determined through Bayesian estimation, specifically using Maximum A Posteriori (MAP) estimation [83]. A significant advantage of the MRF model is its ability to incorporate a priori information locally through factorial potentials [84]. Typically, MRF models generate non-convex energy functions, the minimization of which is essential to accurately identify the most probable segmentation according to the MRF framework [85].

#### 2.5.4. Shortest Path Methods

The shortest path between two vertices in a graph is defined as the path that minimizes the sum of the weights of its constituent edges. This classical problem in graph theory involves finding the most efficient route between two specified vertices, s and t, within a connected weighted graph G. The objective is to identify a path from s to t such that the total weight of the edges along the path is minimized [69].

Several well-established algorithms are commonly employed to find the shortest path in a graph, including Dijkstra’s algorithm [86], which is optimal for graphs with non-negative weights; Bellman–Ford’s algorithm [87], which can handle graphs with negative weights; and Floyd–Warshall’s algorithm [88,89], suitable for computing shortest paths between all pairs of vertices in a weighted graph.

## 3. Deep Learning in Medical Image Segmentation

While traditional segmentation techniques have been widely used in various medical imaging tasks, their limitations, particularly when handling complex and high-dimensional data, have driven the development of more advanced methods. Deep learning, with its capacity to automatically learn features from data, has emerged as a powerful tool to overcome these challenges and enhance segmentation accuracy in complex medical images.

Session 3 provides a comprehensive overview of deep learning techniques, including Convolutional Neural Networks (CNNs), Fully Convolutional Networks (FCNs), U-Net Architecture, Recurrent Neural Networks (RNNs), Generative Adversarial Networks (GANs) and Autoencoders (AEs). It will also explore the application of each of these deep-learning techniques to medical image segmentation. Finally this session will discuss some of the current challenges in deep learning.

### 3.1. Overview of Deep Learning Techniques

Deep learning models are deep artificial neural networks composed of an input layer, an output layer, and multiple hidden layers [90]. Various deep-learning architectures have become dominant in the field, with CNNs serving as the foundation for most segmentation models. Section 3.1 provides an overview of the primary deep-learning techniques used in medical image segmentation.

#### 3.1.1. Convolutional Neural Networks (CNNs)

CNNs [13] are classical models resulting from the integration of deep learning and image processing technologies. As one of the most representative neural networks in deep learning, CNNs have made numerous breakthroughs in image analysis and processing. The predecessor of CNNs is the neocognitron, proposed by Fukushima [91], inspired by Hubel and Wiesel’s [92] hierarchical receptive field model of the visual cortex. LeCun et al. [93] further developed the modern CNN architecture for document recognition (Figure 5), which evolved into LeNet-5, a multilayer artificial neural network capable of classifying handwritten digits.

Figure 5 illustrates the architecture of a typical CNN used for image tasks. The core structure of a CNN consists of three main components: the convolutional layer, the pooling layer, and the fully connected layer [94]. The process begins with an input image, which is passed through multiple convolution layers (represented by the green boxes). In these layers, various convolutional filters are applied to extract key features from the image, such as edges, textures, and patterns. After each convolution, the image is downsampled using pooling layers (blue boxes), which reduce the spatial dimensions of the feature maps, helping to retain essential information while reducing computational complexity. This sequence of convolution and pooling is repeated multiple times to capture increasingly complex and abstract features. Finally, the network flattens the feature maps and passes them through fully connected layers (yellow boxes), where the learned features are combined to make a high-level decision, generating an output, typically in the form of a classification label or prediction.

Most notably, Krizhevsky et al. [95] introduced a classical CNN architecture known as AlexNet, which demonstrated significant improvements over previous approaches in image classification tasks. The success of AlexNet spurred the development of many subsequent models aimed at enhancing performance, with four representative architectures being ZFNet [96], VGGNet [97], GoogleNet [98], and ResNet [99].

A non-linear activation function, such as ReLU (Rectified Linear Unit), is often applied after the convolutional layer to introduce non-linearity, allowing the network to learn complex patterns. In traditional CNN architectures, fully connected layers are used in image classification tasks to map high-dimensional features to output categories.

While traditional CNNs are primarily used for image classification, where the goal is to assign a category label to the entire image, image segmentation requires assigning a label to each pixel, necessitating adjustments to CNN architecture [100]. The fully connected layers typically used for classification are replaced with fully convolutional networks (FCNs) to enable pixel-level predictions [90]. This allows the network to not only classify the entire image but also predict the class of each pixel (e.g., tumor, background, or organ), enabling precise image segmentation.

#### 3.1.2. Fully Convolutional Networks (FCNs)

FCNs are a specialized form of CNNs designed for dense prediction tasks, such as image segmentation. The concept of FCNs was first introduced by Long et al. [15] specifically for image segmentation. Unlike traditional CNNs, which rely on fully connected layers for classification tasks, FCNs remove the fully connected layers and use only convolutional layers, enabling pixel-level predictions. The process begins with standard convolutional layers that extract features from the input image, detecting various patterns from low-level features like edges to more complex shapes in deeper layers. Following convolution, pooling layers reduce the spatial dimensions of the feature maps, effectively downsampling the image while summarizing key features.

In FCNs, fully connected layers are replaced by fully convolutional layers, allowing the network to retain spatial dimensions and predict the category of each pixel (e.g., skin lesion segmentation [101], brain tumor segmentation [102], or retinal blood vessel segmentation [103]) rather than classifying the entire image as one label. After feature extraction and downsampling, FCNs use upsampling techniques such as transposed convolution (deconvolution) or bilinear interpolation to restore feature maps to the original image resolution. This produces a dense output with the same spatial dimensions as the input image, enabling precise pixel-level segmentation [104]. Advanced versions introduce skip connections to combine low-level and high-level features, improving the preservation of boundary details in the segmentation map [94].

As shown in Figure 6, the FCN architecture begins with the input image, which passes through a series of convolutional layers to progressively extract features at different levels of abstraction. These features are then downsampled before being restored to their original resolution through upsampling techniques. The final output is a segmentation map, where each pixel is classified according to its predicted label.

#### 3.1.3. U-Net Architecture

Based on FCNs, Ronneberger et al. [16] developed the U-Net network (Figure 7) for biomedical image segmentation, which has become widely used in the field since its introduction.

Figure 7 illustrates the U-Net architecture, which is widely used for medical image segmentation. The network has a U-shaped structure, with an encoder on the left that captures features through convolutional and max-pooling layers and a decoder on the right that restores the original resolution using up-convolution layers. Skip connections between the encoder and decoder ensure that important high-resolution features are retained, leading to more accurate and detailed segmentation results. The final output is a segmentation map that classifies each pixel in the input image [15]. U-Net’s suitability for medical image segmentation lies in its ability to combine low-level and high-level information, where low-level information enhances accuracy and high-level information helps extract complex features [104].

The improvement of U-Net has become a prominent research focus in medical image segmentation, leading to the development of numerous variants. Cicek et al. [105] introduced a three-dimensional (3D) U-Net model aimed at enriching spatial information for volumetric images while retaining the original strengths of FCN and U-Net. Milletari et al. [106] proposed the V-Net, a U-Net structure adapted for 3D data, which utilizes the Dice coefficient loss function instead of the traditional cross-entropy loss. Zhang et al. [107] introduced Residual U-Net (ResU-Net), where residual connectivity helps mitigate the vanishing gradient problem in deep neural networks, making the network easier to train. Oktay et al. [108] proposed an Attention U-Net, which incorporates an attention mechanism into U-Net, allowing the network to adaptively focus on critical regions in the image while ignoring irrelevant areas. The attention mechanism learns the weights to automatically emphasize pixels crucial to the segmentation task during feature extraction.

#### 3.1.4. Recurrent Neural Networks (RNNs)

RNNs were originally designed to process sequential data, such as text and time-series data, due to their “memory” properties that capture temporal dependencies in input data [109]. However, standard RNNs struggle with long sequential data due to the gradient vanishing problem, which also affects image segmentation tasks. To address this, researchers often use LSTM (Long Short-Term Memory) [110] or GRU (Gated Recurrent Unit) [18], both of which are variants of RNNs that enhance memory capabilities.

LSTM is a specialized RNN structure that captures temporal dependencies by introducing gates—specifically the forget gate, input gate, and output gate—which control the flow of information. This structure allows LSTM to effectively retain long-range dependency information, which is crucial when processing long-distance pixel dependencies in large images. For example, when segmenting complex liver or heart images, distant tissue and boundary information can be maintained effectively by the LSTM network. GRU, a simplified version of LSTM, contains only two gates—the reset gate and the update gate—reducing computational complexity. Despite its simpler structure, GRU performs comparably to LSTM in many tasks and is particularly suitable for resource-limited applications.

#### 3.1.5. Generative Adversarial Networks (GANs)

GANs have shown great promise in various computer vision tasks, including image segmentation. The concept of GANs was first introduced by Goodfellow et al. [19] for synthesizing images from noise. Figure 8 illustrates the structure of GANs, which comprise two primary components: the generator and the discriminator. The generator network takes random noise as input and generates synthetic images that mimic real ones. In contrast, the discriminator network receives both real images and the synthetic images produced by the generator, and its task is to distinguish between the two. The discriminator outputs a prediction of whether the input is “real” or “fake”. During training, the generator continuously improves by creating increasingly realistic images to deceive the discriminator, while the discriminator simultaneously enhances its ability to correctly identify real versus fake images. This adversarial process results in the mutual improvement of both networks, ultimately driving the overall performance of the GAN framework.

Generators in GANs typically optimize two types of losses: adversarial losses and pixel-level losses, such as L2 or cross-entropy [111]. L2 loss [112] measures the difference between predicted pixel values and the ground truth, penalizing large deviations, and is useful for regression tasks like segmentation. Cross-entropy loss [113] is commonly used for classification tasks and evaluates how well the predicted pixel class (e.g., tumor or non-tumor) matches the actual class. By combining these losses, the generator is encouraged to not only produce visually realistic segmentation masks but also ensure that each pixel is correctly labeled, resulting in accurate and consistent outputs.

#### 3.1.6. Autoencoders (AEs)

AEs [114] are unsupervised learning algorithms used for representation learning, dimensionality reduction, and feature extraction. Figure 9 illustrates the generic architecture of AEs. The Autoencoder consists of two main components: the encoder and the decoder. The encoder compresses the input image into a latent representation, which captures the most important features of the input in a lower-dimensional space. This latent representation acts as a compressed code that retains critical information about the input. The decoder then takes this latent representation and reconstructs the input image as closely as possible to its original form. The process allows the Autoencoder to learn efficient representations of the input data, which can be useful for tasks such as image denoising, compression, or feature extraction.

Although originally designed for tasks such as image compression and denoising, the architecture of autoencoders is also well-suited for image segmentation, where the objective is to classify each pixel in an image.

To enhance the performance of basic autoencoders for segmentation tasks, variants such as Convolutional Autoencoders (CAEs) [115], Variational Autoencoders (VAEs) [20], and SegNet [116] have been developed, demonstrating significant potential in image segmentation.

### 3.2. Challenges and Limitations of Deep Learning

Despite the considerable advancements that deep learning has brought to medical image segmentation, it still faces several notable challenges and limitations compared to traditional segmentation techniques. One of the primary constraints is the reliance on large, annotated datasets [117]. Deep learning models, particularly CNNs and GANs, require substantial amounts of labeled training data to effectively learn meaningful representations. However, in the context of medical imaging, obtaining such annotated datasets is both time-intensive and costly, as it necessitates expert input. To mitigate the challenge posed by deep learning’s dependence on large, annotated datasets, several strategies have been developed. One such approach is transfer learning [118], where models pre-trained on large, generalized datasets (e.g., ImageNet [119]) are fine-tuned on smaller, domain-specific medical datasets. This enables the model to leverage knowledge from a broader dataset, reducing the need for extensive labeled data in the specific task of medical image segmentation. Another key strategy is data augmentation [120,121], which involves artificially expanding the dataset by applying transformations such as rotation, flipping, and scaling. This allows the model to be exposed to a more diverse set of inputs without requiring additional labeled examples. Additionally, synthetic data generation using models like GANs can produce realistic medical images along with corresponding annotations, further addressing the scarcity of labeled data [122]. By contrast, traditional methods such as thresholding and edge detection often demand significantly less training data or can be directly applied using predefined rules.

Another critical challenge lies in the high computational demands of deep learning models [123]. Architectures like CNNs, FCNs, and GANs require significant computational power, including high-performance GPUs, to train effectively. This can present a barrier in resource-constrained environments, where traditional methods, with their lower computational overhead, are more readily deployable. Furthermore, deep learning models typically entail prolonged training times, especially when processing large 3D medical images, a challenge that many conventional segmentation approaches do not face. Several key strategies have been widely adopted to address the high computational demands of deep learning models. One approach is model compression [124], which includes techniques such as pruning and quantization. These methods reduce the network size by eliminating redundant parameters and compressing the weights, thereby lowering computational costs without significantly compromising accuracy. Another solution involves using lightweight architectures [125], such as MobileNetV2 [126] and EfficientNet [127], which are specifically designed for efficiency, offering strong performance with reduced computational resource requirements. Additionally, distributed and parallel computing techniques [128], including the use of GPUs [129] and TPUs [130], can significantly accelerate the training process by distributing computations across multiple processors. Collectively, these strategies enable deep learning models to perform complex tasks, such as medical image segmentation, with reduced computational overhead, making them more feasible for real-world applications.

Interpretability is another key limitation of deep learning models [131]. While traditional methods, such as edge-based or region-based segmentation, produce transparent, rule-based outputs, deep learning models are frequently perceived as “black boxes”. Complex models, including AEs and GANs, extract intricate features from the data, yet their decision-making processes are often opaque. This lack of interpretability poses particular challenges in clinical settings, where understanding the rationale behind a segmentation outcome is crucial for medical professionals. Several strategies have emerged to improve transparency and interpretability. One prominent approach is the use of attentional mechanisms [132], which allow the model to focus on the most relevant parts of the input data and provide insights into which areas the model has considered in its decision-making process. Another technique is Layer-Wise Relevance Propagation (LRP) [133], which traces predictions through network layers to understand which input features contribute most to a particular output. In addition, interpretable agent models [134], such as decision trees [135] or linear models [136], can be trained based on the predictions of deep learning models to provide a simpler and more understandable explanation of the decision-making process.

Finally, deep learning models often struggle with generalization when applied to unseen data [137]. Traditional, rule-based methods tend to demonstrate more consistent performance across different datasets, whereas deep learning models are prone to overfitting, particularly when trained on datasets with limited variability [104]. Overfitting is a common challenge in deep learning, and several approaches have been proposed to address it, as discussed in the studies reviewed in Section 3.3. For instance, Rehman et al. [138] effectively employed dropout and regularization techniques in their BU-Net model to mitigate overfitting in brain tumor segmentation, leading to improved performance across different MRI datasets. Similarly, Chen et al. [139] utilized data augmentation and integrated learning methods to enhance the generalization of liver tumor segmentation models. In retinal vascular segmentation, Park et al. [103] introduced an innovative GAN-based adversarial training approach, which reduces overfitting by balancing generators and discriminators, thereby improving segmentation robustness across multiple datasets. These and other methods discussed in Section 3.3 demonstrate that while overfitting remains a challenge, it can be effectively managed with strategies tailored to specific tasks and datasets, ensuring better model generalization.

### 3.3. Applications of Deep Learning in Medical Image Segmentation

In the field of medical image segmentation, deep learning has demonstrated remarkable success in overcoming the challenges associated with diverse anatomical regions and pathological conditions. By automatically extracting complex features from medical images, deep learning has substantially improved segmentation accuracy, particularly in cases where traditional methods falter due to variability in intensity, shape, and texture.

This section focuses on the application of deep learning techniques in the last five years in the segmentation of seven key anatomical regions: brain tumors, breast, liver tumors, lungs, prostate, retinal vessels, and skin lesions. These regions are among the most clinically significant, where precise segmentation is essential for disease detection, treatment planning, and outcome monitoring. For each anatomical area, we will explore how deep learning architectures have been employed to enhance segmentation performance, addressing the specific challenges and complexities unique to each region.

To enable a standardized comparison of the segmentation techniques reviewed in this paper, a consistent set of evaluation metrics, including Dice Similarity Coefficient (DSC), Jaccard Index (IOU), Recall, Accuracy, Sensitivity (SE), and Specificity (SP). These metrics were chosen due to their widespread acceptance in the field of medical image segmentation, allowing for consistent and objective performance evaluation across various methods. They were consistently mentioned for cases in Section 3.3.1, Section 3.3.2, Section 3.3.3, Section 3.3.4, Section 3.3.5, Section 3.3.6 and Section 3.3.7, and Table 3 summarizes the performance of deep learning-based segmentation methods applied to different anatomical regions using publicly available medical imaging datasets. This comparison underscores the strengths and limitations of each technique under a unified evaluation framework, providing clearer insights into their applicability across specific clinical and research scenarios.

#### 3.3.1. Brain Tumour

Brain tumor segmentation involves identifying malignant brain tissue and automatically labeling regions according to tumor type. Accurate segmentation is crucial for preserving healthy tissue while targeting tumor cells during treatment.

To address the issue of small brain tumors being lost during downsampling, Daimary et al. [140] introduced three hybrid CNN models: U-SegNet, Res-SegNet, and Seg-UNet. These models were designed for high-accuracy automatic segmentation of brain tumors from MRI images. Trained and validated on the BraTS dataset, they achieved average accuracies of 91.6%, 93.3%, and 93.1%, respectively, demonstrating superior performance compared to existing CNN models.

To tackle U-Net’s limited memory for high-resolution input images and difficulty retaining local details, Lee et al. [141] proposed a patch-wise U-Net architecture for automatic brain structure segmentation in structural MRI. This method divides MRI slices into non-overlapping patches, which are fed into the network along with ground truth data for training. The patch-wise U-Net achieved an average DSC of 0.93, outperforming conventional U-Net and SegNet-based methods by 3% and 10%, respectively.

Rehman et al. [138] improved segmentation accuracy with the BU-Net model, which incorporated two new modules: Residual Extended Jumps (RES) and Wide Context (WC). The WC module enhances segmented region reconstruction, while the RES module prevents information degradation. BU-Net, evaluated on BraTS 2017 and 2018 datasets, achieved Dice scores of 0.901 for Tumor Core (TC), 0.837 for Whole Tumor (WT), and 0.788 for Enhanced Tumor (ET), showing significant improvements over U-Net, with gains of 7%, 6.6%, and 8.5%, respectively.

To address challenges such as high heterogeneity and low resolution in multimodal MRI images, Tan et al. [142] developed ACU-Net, a multimodal segmentation method that uses deep separable convolutions to enhance computational efficiency. Dense residues and full-size skip connections were added to stabilize loss and improve global feature fusion. Additionally, the active contour model was employed to mitigate noise and edge blurring, achieving Dice scores of 0.9273 for WT, 0.9580 for TC, and 0.8429 for ET on BraTS datasets.

Zhang et al. [143] proposed AGResU-Net, an attention-gated residual U-Net model that integrates residual modules and attention gates into the U-Net architecture. Attention-gated units were added in skip connections to highlight salient features and filter out irrelevant noise. AGResU-Net outperformed U-Net and ResU-Net on benchmarks from BraTS 2017–2019.

Aboussaleh et al. [144] developed Inception U-Det, an improved U-Net architecture inspired by U-Det, which replaces convolutional blocks with inception blocks. This modification reduces dimensionality, shortens execution time, and mitigates the vanishing gradient problem. Inception U-Det achieved Dice scores of 87.9%, 85.5%, and 83.9% on BraTS 2020, 2018, and 2017 datasets, respectively.

To address the limitations of conventional V-Net in multimodal segmentation, Zhang et al. [145] proposed the Multi-Encoder Network (ME-Net), which uses four separate encoders to handle different MRI modalities. ME-Net processes multiple inputs simultaneously, extracting features unique to each modality. It achieved average Dice scores of 0.70249 for ET, 0.88267 for WT, and 0.73864 for TC using the BraTS Challenge validation tool. Similarly, Guan et al. [197] introduced 3D AGSE-VNet, incorporating Squeeze-and-Excite (SE) modules and Attention-Guided Filters (AG) to enhance channel information and suppress noise. This model achieved Dice scores of 0.68 for ET, 0.85 for WT and 0.69 for TC on the BraTS 2020 dataset.

Zhou et al. [146] addressed spatial information loss in traditional convolutional networks with AFPNet, a 3D fully convolutional network that integrates a pyramid of convolutional features for brain tumor segmentation. Tested on the BRATS 2013 dataset, AFPNet achieved Dice scores of 0.68 for ET, 0.86 for WT, and 0.73 for TC. Sun et al. [102] designed a novel 3D fully convolutional model using multipath convolutional layers, which extract features from the input data and fuse them in transposed convolutional layers. Tested on BraTS 2018, the model achieved Dice scores of 0.90 for WT, 0.79 for TC, and 0.77 for ET.

#### 3.3.2. Breast

Breast cancer is one of the most prevalent diseases affecting women globally. To tackle the challenge of automatic segmentation, which is complicated by variations in breast lump size and image features, Byra et al. [147] developed a selective kernel (SK) U-Net convolutional neural network that efficiently segments breast lumps in ultrasound images. The method is based on a U-Net model enhanced with a selective kernel, allowing the network to automatically adjust receptive fields for improved segmentation performance compared to the standard U-Net. The average Dice scores for segmentation on the UDIAT, OASBUD, and BUSI datasets were 0.780, 0.676, and 0.646, respectively.

Piantadosi et al. [148] enhanced their previous U-Net deep convolutional neural network for breast segmentation in MRI, originally proposed in 2018 [198], by incorporating an efficient CNN integration and a voter mechanism to exploit DCE multiplanarity for 3D breast tissue segmentation. The median Dice similarity indices in the DSprivate42 and DSpublic88 datasets were 96.60% (±0.30%) and 95.78% (±0.51%), respectively, with *p* < 0.05 and 100% tumor lesion coverage.

Inspired by the success of U-Net and its variants, Baccouche et al. [149] proposed a new architecture for segmenting breast mass tumor regions of interest (ROIs) by connecting two simple U-Nets, referred to as Connected-U-Nets. This architecture provides two cascaded encoders and decoders that are alternately connected through skip connections. Additionally, it integrates Atrous Spatial Pyramid Pooling (ASPP) to emphasize contextual information. The proposed architecture achieved high Dice scores of 89.52%, 95.28%, and 95.88%, along with Intersection over Union (IoU) scores of 80.02%, 91.03%, and 92.27% on the CBIS-DDSM, INbreast, and private datasets, respectively.

To improve the performance of the U-Net model, Guo et al. [150] developed a fully automated real-time image segmentation and recognition system for breast ultrasound-guided intervention. They analyzed practical scenarios for semantic segmentation of breast ultrasound images using U-Net and added a dropout layer to reduce texture detail redundancy and prevent overfitting. The system achieved a mean Dice coefficient of 90.5% (±0.02) and a mean IoU of 82.7% (±0.02) with an extended training method. This approach facilitates accurate and automatic multi-class recognition of breast ultrasound images.

Addressing issues related to multi-scale and edge blurring in breast lesion segmentation, Li et al. [151] proposed a deep learning architecture called Multi-scale Fusion U-Net (MF U-Net), based on U-Net. The architecture includes a fusion module (WFM) for segmenting irregular breast lesions, a multi-scale dilated convolution module (MDCM) to handle segmentation challenges posed by large-scale variations, and a focal-DSC loss to address class imbalance in breast lesion segmentation. The MF U-Net achieved state-of-the-art segmentation results on the BUSIS dataset, with a recall of 0.9421, precision of 0.9345, FPs/image of 0.0694, DSC of 0.9535, and IoU of 0.9112.

Robin et al. [152] proposed a U-Net-based architecture for tumor region segmentation in histopathological images. The network relies on a fully convolutional design that reduces the need for extensive training data and improves segmentation accuracy. Their method achieved an overall accuracy of 94.2% with a small dataset.

Xue et al. [153] introduced GG-Net, a CNN with a Global Guidance Block (GGB) designed to address CNN’s limited ability to capture long-range dependencies in ultrasound images. The GG-Net aggregates non-local features in the spatial and channel domains, guided by multilayer feature integration, to learn robust non-local contextual information. Additionally, a Breast Lesion Boundary Detection (BD) module was developed within the framework to enhance boundary quality. GG-Net outperformed state-of-the-art segmentation methods in experimental evaluations on two ultrasound breast lesion datasets.

To reduce workload and improve segmentation accuracy in four-dimensional (4D) DCE-MRI, Khaled et al. [154] proposed an automated breast lesion segmentation method based on a U-Net framework. This method uses an ROI-guided, 3D patch-based U-Net previously proposed by Khaled [199] and incorporates residual blocks and an ensemble approach combining three different models. The new method achieved an average DSC of 0.680 (0.802 for major foci) on the TCGA-BRCA dataset, outperforming existing methods on the same dataset.

Ning et al. [155] addressed issues of low-contrast appearance, blurred borders, and shape variations in breast ultrasound (BUS) images by developing a saliency-guided morphology-aware U-Net (SMU-Net). The SMU-Net consists of a primary network, an intermediate stream, and an auxiliary network. The co-learning of foreground and background saliency representations allows the network to utilize rich background texture information to assist in foreground segmentation. Extensive experiments on five public datasets demonstrated the superiority of SMU-Net in both performance and robustness over state-of-the-art deep learning methods.

Zhai et al. [156] proposed a novel asymmetric semi-supervised GAN (ASSGAN) based on the GAN framework, which employs two generators and a discriminator for adversarial learning. The generators supervise each other, producing reliable segmentation prediction masks without the need for labeling, thus facilitating model training using unlabeled cases. ASSGAN outperformed fully supervised methods by 4.16% to 13.94% in IoU across several public BUS datasets (DBUI, OASBUI, SPDBUI) and the custom SDBUI dataset.

#### 3.3.3. Liver Tumor

Liver cancer is the leading cause of cancer-related deaths globally, with early detection via CT offering the potential to prevent millions of deaths annually. However, the fast and accurate extraction of the liver from CT scans remains a significant challenge and a bottleneck for any automated detection system.

Almotairi et al. [157] modified the SegNet deep learning technique to improve tumor segmentation in CT liver scans in DICOM format. By replacing the classification layer with a binary pixel classification layer, the model facilitated the binary segmentation of medical images. Using the 3D-IRCADb-01 dataset for training and testing, the method detected most tumor regions accurately, achieving a classification accuracy exceeding 86%. However, a few false positives remain, which could be mitigated with a false positive filter or by training the model on a larger dataset.

To enhance segmentation accuracy in liver and tumor regions, Budak et al. [158] introduced a new deep learning method utilizing a cascaded encoder–decoder convolutional neural network (CEDCNN) for liver and tumor detection in CT images. The model, tested on the 3DIRCADb dataset, achieved an average DSC of 95.22%, outperforming other existing liver segmentation methods.

Addressing the challenges of fuzzy boundaries and irregular liver shapes, Aghamohammadi et al. [200] proposed a cascaded convolutional neural network (TPCNN) that combines three input images to identify liver and tumor boundaries in abdominal CT images. By applying a Z-Score normalization method and LDOG encoding, the approach improved segmentation accuracy, particularly for fuzzy boundaries. Sabir et al. [159] implemented a deep, dense network (ResU-Net) on CT scans, utilizing the 3D-IRCADb01 dataset to segment the liver and tumor regions. The ResU-Net system achieved a DSC of 0.97% for organ recognition and 0.83% for tumor segmentation, outperforming state-of-the-art methods. Sirco et al. [201] evaluated four liver segmentation methods using ResNet variants (ResNet-18, ResNet-34, ResNet-50, ResNet-101) on 130 CT datasets, finding that ResNet-34 achieved the highest accuracy at 99.2%, while ResNet-101 proved the most efficient, and ResNet-18 was the fastest.

Despite U-Net’s widespread use for medical image segmentation, its encoder-decoder structure can be limited in certain applications. To address this, Tran et al. [160] proposed a new structure for convolutional nodes that integrates dilated convolutions (DC) and a dense structure to enhance learning depth. On the LiTS dataset, this model achieved a DSC of 96.38% for liver segmentation and 73.69% for tumor segmentation, with similar results on the 3DIRCADb dataset.

To leverage the 3D spatial information in medical images, which 2D networks often fail to capture, Chen et al. [139] developed a biplane joint method combining segmentation results from coronal and transverse slices to retain spatial information. Integrated with Deep Residual Attention U-Net (DRAUNet), the method achieved DSC scores of 97.3%, 97.4%, and 96.9% on the LiTS, 3DIRCADb, and Sliver07 datasets, respectively, outperforming other state-of-the-art networks.

To address noise, non-homogeneity, and variability in tumor appearance, Roy et al. [161] introduced an automated liver tumor segmentation method combining masked RCNN for liver segmentation with MSER for tumor identification. The model achieved an average segmentation accuracy of 87.8% for hepatocellular carcinoma (HCC), malignant tumors (excluding HCC), and benign tumors. Similarly, Selvaraj et al. [162] proposed the Convolutional Encoder-Decoder Residual Neural Network (CEDRNN), which uses additive jump-connected feature accumulation to improve learning efficiency and training convergence. The model, tested on the 3DIRCADb dataset, achieved a DSC of 95.2% for liver tumor segmentation.

Wang et al. [163] employed the UNet++ network for initial liver segmentation, reintroducing the segmented liver mask as input for tumor segmentation. By integrating T2-weighted and T1-enhanced sequences, the model achieved DSC scores of 0.612 and 0.687 for liver tumors on the validation and internal test sets, respectively, showing promise for reducing manual annotation time and supporting advanced imaging genomics research.

#### 3.3.4. Lung

Lung cancer is the second most common type of cancer globally, affecting both men and women [202]. According to the World Health Organization (WHO), approximately 1.3 million people succumb to lung cancer each year worldwide [203]. Early diagnosis plays a pivotal role in reducing mortality and improving patient survival rates. Therefore, continuous advancements in intelligent algorithms for the early detection of lung cancer are critical.

Gaál et al. [164] developed a state-of-the-art fully convolutional neural network-integrated with an adversarial critic model for segmenting lungs from chest X-rays. While the adversarial scheme contributed primarily to minor shape refinements, the model achieved an impressive DSC of 97.5% on the JSRT dataset.

To address challenges in lung region delineation within CT scans, Hu et al. [165] applied a Mask R-CNN model in combination with both supervised and unsupervised machine learning methods, including Bayesian models, Support Vector Machines (SVMs), K-means, and Gaussian Mixture Models (GMMs). The Mask R-CNN, coupled with a K-means kernel, delivered the best segmentation results, achieving an accuracy of 97.68 ± 3.42% with an average runtime of 11.2 s.

Khanna et al. [166] tackled the issue of shallower networks’ limited ability to extract discriminative features by introducing a deep learning-based architecture known as Residual U-Net, coupled with a false-positive removal algorithm for lung CT segmentation. The architecture, which incorporates residual blocks and various data enhancement techniques to improve generalization, demonstrated superior performance. The proposed method achieved DSC values of 98.63%, 99.62%, and 98.68% on the LUNA16, VESSEL12, and HUG-ILD datasets, respectively.

Munawar et al. [167] were pioneers in proposing a GAN-based model for lung segmentation from chest X-rays. The model, which consists of a generator and multiple discriminators akin to the U-Net architecture, uses adversarial loss to refine the segmentation masks. The model demonstrated strong performance across three CXR datasets, yielding a DSC of 0.9740 and an Intersection over Union (IoU) score of 0.943.

Recognizing the influence of surrounding tissues and the heterogeneity of lung nodules, Xiao et al. [168] proposed a segmentation method for CT images based on a 3D-UNet and Res2Net, introducing the 3D-Res2UNet architecture. This novel design addresses challenges such as gradient vanishing and explosion, leading to improved detection and segmentation accuracy. Tested on the LUNA16 public dataset, the network achieved a DSC of 95.30% and a recall rate of 99.1%, underscoring its efficacy in lung nodule segmentation.

To enhance the accuracy and efficiency of lung CT image segmentation, Chen et al. [169] proposed the DC-U-Net model. This architecture incorporates a cavity convolution structure to expand the model’s receptive field without increasing the number of parameters and includes 1 × 1 convolution layers to enhance non-linear representation. The DC-U-Net achieved an IoU of 0.9627 and a DSC of 0.9743, outperforming the traditional U-Net model.

Addressing the lack of binary masks in current lung CT databases, Jalali et al. [170] developed a semi-automatic approach for generating corresponding masks. Ground truth was extracted via morphological operations and manual refinements before being processed through a modified U-Net architecture. By replacing the encoder with a pre-trained ResNet-34 and incorporating BConvLSTM for high-level feature integration, the proposed Res BCDU-Net achieved a DSC of 97.31% on the LIDC-IDRI database.

To streamline the segmentation process, Tan et al. [171] introduced a GAN-based segmentation framework (LGAN) for lung CT images. The model utilizes a generator and discriminator network, with the generator predicting lung masks and the discriminator refining these predictions via Earth Mover’s (EM) distance. The LGAN framework significantly outperformed existing methods on the LIDC-IDRI dataset, achieving a Dice score of 0.985.

Liu et al. [172] addressed the issue of gradient instability by employing a pre-trained EfficientNet-B4 as the encoder, along with residual blocks and a LeakyReLU activation function in the decoder. This approach improved both the DSC and Jaccard index by approximately 2.5% and 6%, respectively, across two benchmark lung segmentation datasets and by 5% and 9% on a private dataset. The method demonstrated robustness and a low standard deviation in segmentation outcomes.

#### 3.3.5. Prostate

Prostate cancer is among the most prevalent cancers in men, and precise segmentation of the prostate is a critical step in optimizing radiation dose delivery to the tumor while minimizing damage to adjacent healthy organs.

Astono et al. [173] optimized a basic two-dimensional (2D) U-Net model for prostate segmentation using a private dataset of T2-weighted MR images. Their model demonstrated superior upsampling performance, employing interpolation and convolution in place of transposed convolution. Notably, the integration of feature mapping and skip connections was advantageous primarily in serial operations. Additionally, average pooling significantly outperformed other pooling methods, such as max, RMS, and L2. The inclusion of a batch normalization layer prior to activation further enhanced model performance, leading to median and average DSC improvements of approximately 6% and 7%, respectively, over traditional segmentation methods on the public dataset PROMISE12.

Recognizing the heterogeneous and inconsistent pixel representation near the prostate boundary, Hambarde et al. [174] proposed a radiomics-based, deeply supervised U-Net for the segmentation of both the prostate capsule and lesions in axial T2W MR images. The framework was trained using the Stochastic Gradient Descent (SGD) algorithm, paired with the Adam optimizer, and included additional hyperparameters. Efficient segmentation of prostate lesions at all cancer stages (T1–T4) was achieved, with average DSCs of 0.8958 for prostate capsule segmentation and 0.9176 for prostate lesion segmentation.

Ushinsky et al. [175] enhanced traditional CNN techniques by reconfiguring a standard 2D U-Net into a hybrid 3D–2D U-Net architecture. This hybrid model was trained on 3D images and then employed for segmentation on 2D images. Modifications to the downsampling layers, including 3D convolution and corrected linear unit activation, yielded a five-fold cross-validation paradigm for training and validation. The hybrid 3D–2D U-Net achieved an average DSC of 0.898 and a Pearson correlation coefficient of 0.974 for prostate volume estimation.

To develop a model with high computational accuracy and fast convergence, Chen et al. [176] introduced a 3D AlexNet method for the automatic segmentation of prostate cancer in MRI images. By incorporating the PReLU activation function, the effect of varying convolutional kernel sizes on recognition accuracy was evaluated. The model’s global pooling mechanism compared global max pooling and global average pooling to identify the most effective approach. Experimental results indicated that the model achieved a DSC of 0.9768.

Addressing the limitations of conventional FCNs, which often result in non-smooth segmentation boundaries, He et al. [177] proposed a two-stage framework for prostate segmentation from raw CT images. The first stage involved rapid localization of the prostate region using a lightweight network on downsampled CT images. The second stage employed a multitasking UNet, termed MetricUNet-HCR, which combined segmentation information with voxel-level feature relations to generate refined segmentation maps. Although initial results showed a DSC of 0.9, He et al. [178] later introduced a hierarchical fusion U-Net (HF-UNet) to overcome data fitting limitations in traditional multi-task networks. HF-UNet utilized two complementary branches with attention-based task-consistency learning blocks, resulting in superior performance compared to existing multi-task networks.

Given the substantial variability in image appearance, anisotropic spatial resolution, and imaging disturbances, Jin et al. [179] proposed an automated segmentation method for prostate MRI data using bicubic interpolation and a modified 3D V-Net named 3D PBV-Net. The method utilized double cubic interpolation to pre-process MRI data, addressing the low-frequency components in prostate imaging. The 3D PBV-Net demonstrated strong segmentation performance, achieving average accuracies of 97.65% and 98.29% and DSC scores of 0.9613 and 0.9765 on the PROMISE12 and TPHOH datasets, respectively.

To further enhance segmentation performance across varying locations in prostate MRI images, Wang et al. [180] developed SegDGAN, an end-to-end architecture based on the GNN model. This approach combined densely connected blocks in a fully convolutional generator network with a multi-scale, multi-level convolutional discriminator network. SegDGAN outperformed U-Net, FCN, and SegAN, achieving the highest DSC values and the lowest VOE, ASD, and HD values—86.24%, 23.60%, 1.02 mm, and 7.57 mm, respectively—on the PROMISE12 public dataset.

#### 3.3.6. Retinal Vessel

The retinal vasculature plays a critical role in diagnosing many ocular diseases, making accurate segmentation essential for effective diagnosis.

To overcome the limitations of existing FCNNs that involve a large number of hyperparameters and increased end-to-end training time due to their decoder structures, Khan et al. [181] proposed a novel supervised deep learning-based approach for retinal vessel segmentation. This method extends a variant of FCNNs by sharing the indexes obtained from maximal pooling in the encoder at each stage with the decoder, thereby enhancing the resolution of the feature map. This significantly reduces the number of tunable hyperparameters and computational overhead during training and testing. A key innovation is the use of external jump connections to convey preserved low-level semantic edge information, which improves the detection of fine blood vessels in retinal fundus images. Experimental results demonstrate reduced computational complexity, memory requirements, and high robustness in microvessel segmentation.

Most supervised and unsupervised methods suffer from a lack of segmentation robustness, leading to spurious or fine branches due to suboptimal loss function optimization. To address this issue, Park et al. [103] introduced a novel conditional generative adversarial network, M-GAN, to achieve accurate retinal vessel segmentation by balancing the loss through stacked deep, fully convolutional networks. M-GAN features an M-shaped architecture with a generator that includes long-term residual connections between down-sampled and up-sampled network layers. It also incorporates a binary cross-entropy loss function and a pseudo-negative loss function to enhance training efficiency and segmentation robustness. The proposed model achieves balanced precision and recall through the FN loss function, yielding the highest IoU and F1 scores across multiple datasets.

Chala et al. [182] proposed an end-to-end approach based on a multi-encoder-decoder architecture that learns features from CNN models to produce retinal vessel segmentation. Their model includes two encoder units (RGB and green), a decoder, and a module for progressive dimensionality reduction. Each encoder consists of sub-CNNs with three convolutional layers. The model was tested on the DRIVE and STARE datasets, achieving F1 scores of 0.8321, accuracy of 0.9716, sensitivity of 0.8214, specificity of 0.9860, and precision of 0.8466.

To improve the interpretability and reduce the complexity of U-Net, Guo et al. [183] introduced a spatial attention module within a lightweight network model called Spatial Attention U-Net (SA-UNet). The spatial attention module refines feature maps by multiplying the attention graph with the input feature map. Additionally, structured dropout convolutional blocks replace U-Net’s original blocks to prevent overfitting. The model achieved state-of-the-art performance on both the DRIVE and CHASE_DB1 datasets. Similarly, Li et al. [184] proposed a CNN with an attention mechanism to improve retinal vessel segmentation, reducing network parameters while maintaining performance across five common datasets. The attention modules operate similarly to conventional convolutional layers, allowing for the addition of multiple attention modules.

Addressing the difficulty of segmenting vascular structures from low-contrast, blurred retinal images, Zhang et al. [185] developed Pyramid U-Net for accurate retinal vessel segmentation. The model employs a Pyramid Scale Aggregation Block (PSAB) in both the encoder and decoder to aggregate multiscale features, improving capillary detection. Optimizations in the encoder (pyramid input enhancement) and decoder (deep pyramid supervision) further enhance performance. Extensive evaluations show that Pyramid U-Net surpasses state-of-the-art methods on the DRIVE and CHASE-DB1 datasets.

Dong et al. [186] tackled the limitations of simple skip connections between the encoder and decoder, which can degrade accuracy, by introducing Cascaded Residual Attention U-Net (CRAUNet). CRAUNet incorporates DropBlock into the base residual module, effectively reducing overfitting and enhancing segmentation. The model also includes Scale Fusion Channel Attention (MFCA) to leverage multiscale information. CRAUNet demonstrated superior performance on both the DRIVE and CHASE_DB1 datasets, achieving AUC scores of 0.9830 and 0.9865, respectively. Additionally, Liu et al. [187] proposed ResDO-UNet, an encoder-decoder framework with a residual DO-conv (ResDO-conv) network and pooled fusion block (PFB) to improve feature extraction and mitigate information loss. Their model also employs an attention fusion block (AFB) to enhance multi-scale feature expression. Tested on DRIVE, STARE, and CHASE_DB1, ResDO-UNet achieved state-of-the-art detection performance.

Liu et al. [188] proposed Wave-Net, a lightweight segmentation network designed for accurate retinal blood vessel segmentation, specifically addressing thin blood vessel segmentation. To mitigate semantic information loss and enhance microstructure details, the model introduces a Detail Enhancement and Denoising Block (DED), replacing the original U-Net’s simple jump connections. A Multi-Scale Feature Fusion Block (MFF) further improves segmentation accuracy by fusing cross-scale contexts. Experiments show that Wave-Net outperforms advanced segmentation methods, particularly in fine vessel segmentation, while maintaining a lightweight architecture.

#### 3.3.7. Skin Lesion

Skin lesion segmentation is a crucial yet challenging task in the computer-aided diagnosis of dermoscopic images.

To address the challenge of varying scales of view and lesion regions in dermoscopic images, Lei et al. [189] designed a novel GAN with dual discriminators for skin lesion segmentation. The network comprises a segmentation module based on skip connections and a dense convolutional U-Net (UNet-SCDC) combined with a dual discriminator (DD) module. In the UNet-SCDC module, both skip connections and dense inflated convolutional blocks are employed to capture discriminative feature representations and retain fine-grained information for precise segmentation. The DD module enhances the discriminator’s ability to improve decision-making by using two independent but complementary discriminators, which optimizes the generative module more effectively. Extensive experiments on the ISIC Skin Lesion Challenge datasets (2017 and 2018) demonstrated superior segmentation performance compared to state-of-the-art methods.

In cutaneous melanoma detection, the segmentation phase is particularly challenging due to misleading factors such as lesion color, edge variations, hair, markers, frame artifacts, and size inconsistencies. To tackle these issues, Öztürk et al. [190] proposed an efficient segmentation method that leverages spatial information to support the residual structure of the FCN architecture. The improved FCN (iFCN) processes full-resolution skin lesion images without pre-processing or post-processing, effectively determining lesion centers and refining edge details. Tested on the IEEE ISBI 2017 Challenge and PH2 datasets, the method achieved average accuracies of 95.30% and 96.2% and DSC scores of 88.64% and 93.02%, respectively.

To counter the issue of decreasing spatial feature map resolution across network layers, Xie et al. [191] proposed a novel CNN architecture for skin lesion segmentation designed to generate high-resolution feature maps for more accurate boundary delineation. The architecture integrates a High-Resolution Feature Block (HRFB) consisting of primary, spatial attention, and channel attention branches, whose outputs are fused to produce robust features with enhanced detail. Experiments on the ISBI 2016 and 2017 datasets, as well as the PH2 dataset, yielded Jaccard indices of 0.783, 0.858, and 0.857, respectively.

To address variations in dermoscopic image properties caused by fluorescence and luminance inhomogeneity, Zafar et al. [192] developed an automated lesion boundary segmentation method combining U-Net and ResNet architectures, termed Res-Unet. This method integrates an advanced dehairing algorithm that effectively removes hair structures, significantly enhancing segmentation accuracy. Tested on the ISIC-2017 and PH datasets, Res-Unet achieved Jaccard indices of 0.772 and 0.854, respectively.

To implement an automatic classifier capable of distinguishing between various types of nevi without human intervention, Thurnhofer-Hemsi et al. [193] proposed an end-to-end system that employs a deep neural network for classifying dermatological disorders. The system leverages transfer learning across five state-of-the-art CNN architectures to create both general and hierarchical classifiers capable of recognizing seven types of nevi. Trained on the large HAM10000 dermoscopic image dataset and enhanced with data augmentation techniques, the DenseNet201 model achieved high classification accuracy, F-measure, and low false-negative rates.

Recognizing the limitations of conventional network structures in modeling multi-scale objects with significant texture and shape variations, Alahmadi [194] introduced a multi-scale attention U-Net (MSAU-Net) for skin lesion segmentation. MSAU-Net improves upon the standard U-Net by incorporating an attention mechanism at the bottleneck layer to simulate hierarchical representations. The attention module aggregates multi-level features in a non-linear manner, selectively tuning representative features. Furthermore, a bidirectional convolutional long short-term memory (BDC-LSTM) structure is employed to capture discriminative features while suppressing irrelevant ones. MSAU-Net achieved a DSC of 0.9377 on the PH2 dataset.

Anand et al. [195] proposed an improved U-Net architecture for accurate and automatic dermoscopic image segmentation by modifying the feature map dimensionality. The architecture uses pixel-wise classification to locate and distinguish boundaries, ensuring that input and output share the same size. The encoder applies convolutional and max-pooling layers, while the decoder uses transposed convolutional and standard convolutional layers. Experiments on the PH2 dataset, using 8, 18, and 32 different batch sizes with Adam, Adadelta, and SGD optimizers over 25, 50, 75, and 100 epochs, showed the model performed best with a batch size of 8, Adam optimizer, and 75 epochs, achieving 96.27% accuracy, a Jaccard index of 96.35%, and a DSC of 89.01%.

To segment lesions with irregular shapes, low contrast, or blurred boundaries, Dai et al. [196] proposed a multi-scale residual encoding and decoding network (Ms RED). The network employs a Multiscale Residual Coding Fusion Module (MsR-EFM) in the encoder and a Multiscale Residual Decoding Fusion Module (MsR-DFM) in the decoder to adaptively fuse multiscale features. Additionally, a novel multi-resolution, multi-channel feature fusion module (M2F2) is introduced to replace conventional convolutional layers in both the encoder and decoder. For the first time in medical image segmentation, a Soft-pool module is applied to retain more useful information during downsampling, resulting in improved segmentation performance. Experimental results consistently showed that Ms RED significantly outperforms its counterparts across five commonly used evaluation criteria, requiring fewer labeled samples and yielding faster convergence during training.

## 4. Integration of Deep Learning with Traditional Techniques

The combination of traditional segmentation techniques, such as thresholding, edge detection, and region-based methods, with deep learning has proven to be an effective approach to addressing the challenges of medical image segmentation. Traditional methods offer efficiency and interpretability but often struggle with complex or noisy images. Deep learning models, on the other hand, excel at capturing intricate patterns from raw data but require significant computational resources and large labeled datasets. Table 4 presents a detailed comparison of five traditional and six deep learning-based segmentation techniques, outlining their respective advantages, limitations, and best-suited applications in medical image segmentation.

By integrating these two approaches, hybrid models leverage the strengths of each: traditional methods provide a fast, interpretable starting point, while deep learning refines the segmentation to enhance accuracy and robustness. This integration is particularly useful in cases where traditional techniques fall short due to weak edges or intensity variations. Session 4 will explore the practical applications of combining each traditional segmentation technique with deep learning to improve performance in medical image segmentation tasks.

### 4.1. Combining Thresholding Techniques with Deep Learning

Chen et al. [232] utilized a deep CNN for lung X-ray segmentation of contrast-enhanced binarized images. Initially, a contrast enhancement technique specifically designed for chest X-ray (CXR) images was applied, followed by adaptive image thresholding to binarize the images, effectively separating the foreground from the background. The thresholded images were then fed into a CNN-based architecture for segmentation. Experimental results demonstrated that the proposed preprocessing approach was effective across various CNN architectures, achieving segmentation accuracy comparable to state-of-the-art methods. Additionally, this method significantly accelerated model training by up to 20.74% and reduced the storage space required for the CXR dataset by 94.6%.

Isunuri and Kakarla [233] proposed an optimized U-Net model that employs adaptive thresholding as a post-processing step. The adaptive threshold is calculated as the average threshold of the training dataset, designed to retain tumor pixels. During training, the U-Net model is optimized with respect to weights and thresholds specific to the dataset. In the testing phase, the test image is convolved using the optimized U-Net, and adaptive thresholding is applied to generate the final tumor segments. This model demonstrated competitive performance in terms of precision and accuracy, particularly when compared to ResNet34.

Liu et al. [234] introduced a full-resolution network (FR-UNet) coupled with a dual-thresholding iteration (DTI) algorithm for segmentation in retinal vascular and coronary angiography images. In FR-UNet, a feature aggregation module integrates multi-scale feature maps from neighboring stages, enhancing high-level contextual information. The network’s modified residual block continuously learns multi-resolution representations to deliver pixel-level accuracy in prediction maps. The DTI algorithm further improves vascular segmentation by accurately extracting weak vascular pixels, thereby enhancing vascular connectivity. Experimental results showed that FR-UNet combined with DTI outperformed state-of-the-art methods, achieving the highest sensitivity, AUC, F1, and IoU scores on most datasets, with fewer parameters and significantly improved sensitivity.

Reddy et al. [235] proposed an efficient multilevel thresholding scheme for cardiac image segmentation using a GAN. The GAN introduces noise to the images, simulating real-time scenarios, and subsequently, both the original and generated images are segmented through multilevel thresholding to identify regions of interest. This segmentation process aids in predicting the accurate attack rate by considering multiple factors. The method achieved an accuracy of 97.33% and a true positive rate of 90.97%.

### 4.2. Combining Edge-Based Methods with Deep Learning

To address the dynamic topological correlation between feature maps in deep learning frameworks, which often hampers the efficient capture of channel representations, Li et al. [236] proposed a dual-encoder-based dynamic channel map convolutional network with edge enhancement (DE-DCGCN-EE) for retinal vessel segmentation. The model introduces several key innovations. First, a dual encoder is integrated with edge detection to preserve vessel edges during downsampling. Second, a dynamic channel map convolutional network maps image channels into a topological space, synthesizing the features of each channel on this topological map, thereby addressing the under-utilization of channel information. Finally, an edge enhancement module is employed to fuse edge and spatial features within the dual encoder, significantly improving the accuracy of fine vessel segmentation. Experimental results on five retinal image datasets demonstrate that DE-DCGCN-EE outperforms state-of-the-art methods in segmentation accuracy.

Shi et al. [237] proposed a multi-channel CNN-based fuzzy active contour model for medical image segmentation. In this approach, medical images are first meshed and labeled using the superpixel SLIC method. These gridded images are then used as a dataset to train the multi-channel CNN, which segments the hyperpixels at organ boundaries. The seed points of these hyperpixels are connected to form a rough segmentation boundary. Using this rough boundary as the initial contour, the final organ boundaries are determined by applying a fuzzy active contour segmentation technique. Comparative experimental results demonstrate that this method provides superior segmentation robustness and effectiveness.

Wang et al. [238] introduced an iterative edge attention network (EANet) for medical image segmentation, incorporating several innovative components. The Dynamic Scale Sensing Context (DSC) module dynamically adjusts the receptive field to efficiently capture multi-scale contextual information. The Edge Attention Preservation (EAP) module focuses on eliminating noise, enabling the edge flow to concentrate on boundary-related information. Additionally, the Multilevel Pairwise Regression (MPR) module combines complementary edge and region information to refine ambiguous structures, facilitating better representation learning and more accurate saliency maps. Extensive experiments across four challenging medical segmentation tasks—including lung nodule segmentation, COVID-19 infection segmentation, lung segmentation, and thyroid nodule segmentation—revealed that EANet achieves superior performance compared to state-of-the-art methods.

Allah et al. [239] developed the Edge U-Net model, a deep convolutional neural network (DCNN) for brain tumor segmentation. The architecture, inspired by U-Net, incorporates an encoder-decoder structure designed to enhance tumor localization by integrating boundary-dependent MRI data with primary brain MRI data. In the decoder stage, boundary-dependent information from MRIs at different scales is merged with neighboring contextual information. The model introduces a novel loss function that incorporates boundary information, improving the learning process and enhancing segmentation accuracy. Experimental results indicate that the Edge U-Net model achieves high Dice score values, demonstrating strong brain tissue differentiation performance. Specifically, the Dice scores were 88.8% for meningiomas, 91.76% for gliomas, and 87.28% for pituitary tumors, comparing favorably to state-of-the-art models.

### 4.3. Combining Region-Based Methods with Deep Learning

Liu et al. [240] introduced a deep U-Net architecture that incorporates superpixel region merging for edge enhancement, optimizing segmentation performance. Unlike traditional deep learning-based segmentation, this model evolves segmentation from hyperpixel region merging through U-Net training, offering rich semantic information beyond mere grey similarity. Additionally, a bilateral filtering module is applied at the network’s input to reduce external noise and improve soft tissue contrast along pathological edges. To prevent overfitting and enhance sensitivity to model parameters, a normalization layer is inserted after each convolution layer at varying feature scales. The model was validated across lung CT, brain MR, and coronary CT datasets, with cross-validation and hyperpixel methods confirming its effectiveness. Empirical results indicated that a four-layer network offered an optimal balance between precision, recall, F-measure, and runtime efficiency.

Ren et al. [241] proposed ReGANS, a semi-automatic region-growing algorithm for lung nodule segmentation. This approach builds on the traditional Region-growing method by optimizing pixel growth conditions, allowing for automated execution using EP, AvgS, and initial coordinate points. This improvement addresses the limitation of the original method, which required manual pixel growth range settings that often led to inaccurate segmentation. ReGANS enables rapid and accurate segmentation of lung nodules in CT images based on initial manual points.

Khan et al. [242] developed a lightweight CNN model architecture paired with watershed-based region growth segmentation for chest X-rays. This approach utilizes watershed-based segmentation to isolate regions of interest within the images. The lightweight CNN, consisting of just seven convolutional layers, requires significantly less computational power compared to previous state-of-the-art models. Its efficiency makes it suitable for practical applications, including multimodal COVID-19 detection using both X-ray and CT scan images.

### 4.4. Combining Clustering-Based Methods with Deep Learning

Nithya et al. [243] employed artificial neural networks (ANNs) and multi-kernel K-means clustering for kidney disease detection and segmentation in ultrasound images. A median filter was first used to remove noise from the images. Following noise removal, Grey Level Co-occurrence Matrix (GLCM) features were extracted from each image. To improve classification accuracy and reduce complexity, the Crow Search Optimization Algorithm (CSOA) was applied to select the most important features. These selected features were then used by the ANN to classify the images as normal or abnormal. If classified as abnormal, the image proceeded to the segmentation stage, where the multi-kernel K-means algorithm was applied. Experimental results indicated that the system, which combined linear and quadratic segmentation, achieved a maximum accuracy of 99.61%, outperforming other methods.

Khan et al. [244] utilized K-means clustering and deep learning, along with synthetic data augmentation, for brain tumor segmentation and classification. The approach involved converting each input MRI modality into slices, followed by statistical normalization of intensities. K-means clustering was used to segment brain tumors and identify regions of interest (ROIs) for accurate feature extraction. The brain tumors were then classified as either benign or malignant using a fine-tuned VGG-19 CNN model trained with synthetic data augmentation techniques. The results demonstrated superior accuracy compared to previous methods.

Nawaz et al. [245] proposed a deep learning approach for early-stage skin melanoma segmentation, combining Faster Region-Based Convolutional Neural Network (RCNN) with Fuzzy K-means clustering (FKM). The method first preprocesses the dataset images to address noise and illumination issues and enhance visual information. Next, Faster RCNN was applied to extract a fixed-length feature vector. FKM was then used to segment melanoma-affected skin regions with varying sizes and boundaries. The results showed that the proposed method outperformed state-of-the-art techniques, achieving average accuracies of 95.40%, 93.1%, and 95.6% on the ISIC-2016, ISIC-2017, and PH2 datasets, respectively, demonstrating its robustness for skin lesion identification and segmentation.

Fooladi et al. [246] introduced a segmentation method for brain tumors, combining CNNs with Fuzzy K-means clustering. The method involved three stages: image preprocessing to reduce computational complexity, attribute extraction and selection, and segmentation. First, the images were preprocessed using an adaptive filter and wavelet transform to mitigate noise and reduce complexity. Next, a deep neural network was employed for feature extraction, followed by Fuzzy K-means clustering to segment the tumor regions. Validation on the BRATS dataset yielded an accuracy of 98.64%, with 100% sensitivity and 99% specificity, demonstrating the method’s efficacy.

### 4.5. Combining Graph-Based Methods with Deep Learning

Li et al. [247] introduced a fully automated method based on a graph-cutting framework, where a multiscale convolutional neural network (MS-CNN) is employed to learn the potentials of a graph on the surface mesh of the left atrium (LA). The MS-CNN efficiently integrates local and global texture information, significantly enhancing the segmentation accuracy of the graph-cutting method. Further improvements are achieved by balancing the contributions between the t-link and n-link weights of the graph. The proposed approach achieves an average accuracy of 0.856 ± 0.033 and an average Dice score of 0.702 ± 0.071 for LA scar quantification.

Mishra et al. [248] developed a graph-based algorithm combined with deep learning-derived data for automatic retinal layer segmentation. The method utilizes a shortest path framework along with U-Net to enhance performance. The algorithm achieves sub-pixel accuracy for vitreous warts, reticular pseudodrusen (RPDs), and 11 other retinal layers, demonstrating high robustness and precision across this dataset.

Zhang et al. [249] proposed a novel adversarial method based on convolutional neural networks (ARPM-net), enhanced by an MRF for segmentation of the prostate and organs at risk in pelvic CT images. The segmentation network integrates an MRF block into a modified multi-residual U-Net. The discriminator processes the product of the original CT and the predicted or true value, classifying the input as false or true. The segmentation and discriminator networks can be jointly trained or fine-tuned, with adversarial training improving the accuracy of segmentation. ARPM-net achieves state-of-the-art results in segmenting multiple organs in CT images, offering significant potential for routine pelvic cancer radiation treatment planning.

## 5. Discussion and Conclusions

This review has examined traditional, deep learning-based, and hybrid approaches to medical image segmentation, highlighting their respective strengths and limitations. Traditional methods, such as thresholding, edge detection, and region-based techniques, have long been valued for their computational simplicity and interpretability. These methods are efficient and easy to deploy, particularly in resource-constrained environments with limited computational power and smaller datasets. However, their reliance on predefined rules limits their ability to manage the complexity of modern medical images, especially in noisy or heterogeneous clinical datasets. This underscores the need for more adaptive approaches.

Deep learning has transformed medical image segmentation by enabling automatic feature extraction and learning directly from raw medical images, often surpassing traditional methods in accuracy. Techniques such as CNNs, U-Nets, FCNs, RNNs, GANs, and AEs have proven highly effective for segmenting complex anatomical structures, including brain tumors, liver tumors, lungs, prostate glands, and more. These models can manage both high-level and low-level features, making them particularly suitable for medical images, which often contain irregular shapes, weak boundaries, and varying intensity levels.

Despite these advantages, deep learning models present significant challenges, particularly in their high computational demands and need for large, annotated datasets. Current strategies, including model optimization, transfer learning, and semi-supervised learning, offer viable solutions to mitigate these challenges. For instance, transfer learning allows models pre-trained on large datasets to be fine-tuned for specific medical applications, reducing the amount of required data. Additionally, model compression techniques, such as pruning, quantization, and knowledge distillation, make models more lightweight and computationally feasible for clinical deployment. However, further research is needed to simplify and adapt these solutions for real-world use, particularly in hospitals with limited computational resources.

The issue of interpretability remains critical, especially when deep learning models are used in high-stakes healthcare decisions. Clinicians need to understand how these models reach their conclusions to ensure trust and reliability in the decision-making process. Explainable AI (XAI) techniques, such as attention maps or heat maps, can enhance the transparency of deep learning models by highlighting the regions of the image that contribute most to the model’s predictions. For example, saliency maps in GAN or CNN-based models can provide insights into the most relevant regions for tumor segmentation. These features increase confidence in model outputs and align them with clinical needs, where interpretability is essential for informed decision-making.

Hybrid approaches, which combine traditional segmentation techniques with deep learning, show great promise. These methods often leverage traditional techniques as pre- or post-processing steps, enabling deep learning models to focus on the more complex aspects of segmentation. For instance, edge detection or region-growing methods can help delineate boundaries before applying deep learning algorithms, significantly improving segmentation accuracy in cases where deep learning models alone may struggle, such as with weak edges or variable intensities. Traditional methods can also be used as post-processing tools to refine deep learning outputs, smoothing boundaries or enhancing anatomical coherence. Although hybrid models currently represent a smaller portion of research, they offer a compelling solution to the limitations of both traditional and deep learning methods. Future research should focus on optimizing these hybrid techniques, particularly for clinical environments where both adaptability and computational efficiency are crucial.

Looking ahead, developing resource-efficient deep learning models will be a key direction for future research. Clinical environments often have limited access to high-performance computing, making it necessary to create models that can operate on lower-specification hardware. Techniques such as federated learning, which allows multiple institutions to collaborate on model training without sharing sensitive patient data, can help reduce data burdens and improve generalization across diverse patient populations. Cloud-based deep learning solutions can also alleviate computational requirements, enabling smaller hospitals and clinics to access advanced AI tools without requiring significant on-premises infrastructure.

It is evident that hybrid models, integrating traditional and deep learning-based approaches, will play a central role in advancing medical image segmentation. However, making these models more accessible and interpretable in real-world clinical settings should be a top priority. This includes embedding interpretability as a core feature and addressing concerns about the reliability of deep learning outputs, especially when deployed in varied and unpredictable clinical environments. Validating models across multi-institutional datasets will be essential to ensure robustness and generalizability across different imaging modalities, patient populations, and clinical contexts.

In conclusion, while deep learning has driven significant progress in medical image segmentation, its future success will depend on overcoming challenges related to interpretability, computational efficiency, and data availability. Hybrid approaches, which combine the strengths of both traditional and deep learning techniques, provide a pathway to more adaptable, accurate, and clinically feasible solutions. As the field evolves, it is essential that researchers focus on these areas to fully realize the potential of AI in medical imaging and ensure its benefits are translated into everyday clinical practice.

## Figures and Tables

**Figure 1 bioengineering-11-01034-f001:**
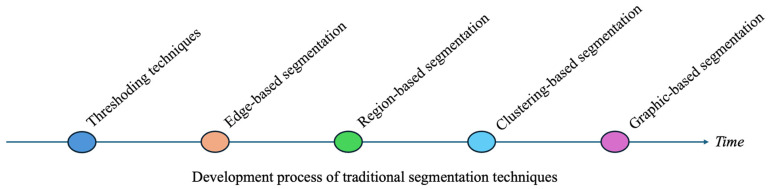
Overview of traditional segmentation techniques in Session 2.

**Figure 2 bioengineering-11-01034-f002:**
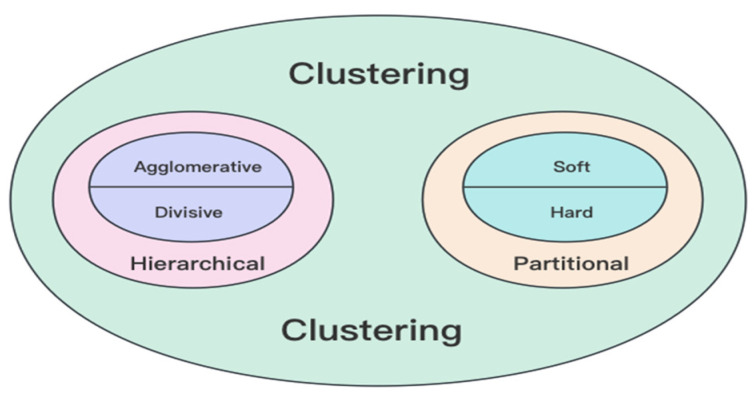
Classification of clustering-based image segmentation methods.

**Figure 3 bioengineering-11-01034-f003:**
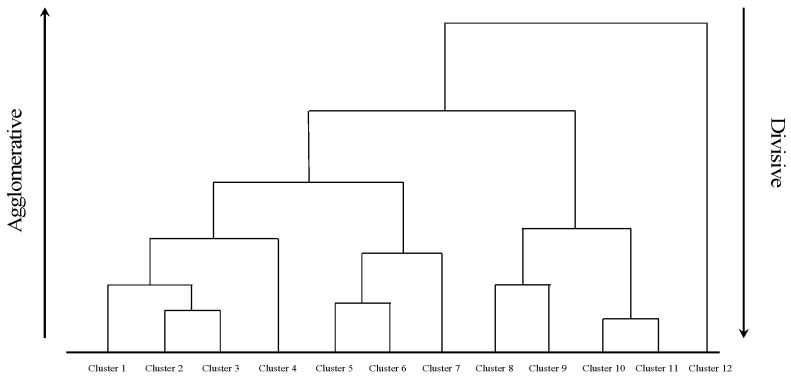
Hierarchical clustering dendrogram.

**Figure 4 bioengineering-11-01034-f004:**
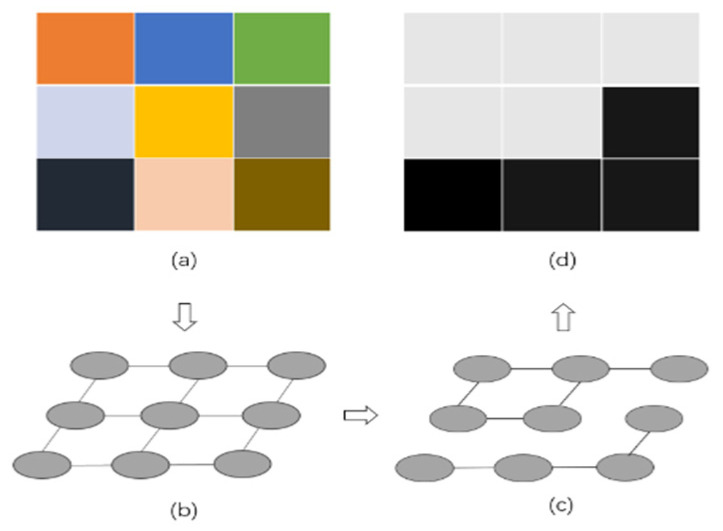
Schematic representation of the relationship between image segmentation and graph partitioning. (**a**) Image; (**b**) graph; (**c**) graph partitioning; (**d**) image segmentation.

**Figure 5 bioengineering-11-01034-f005:**
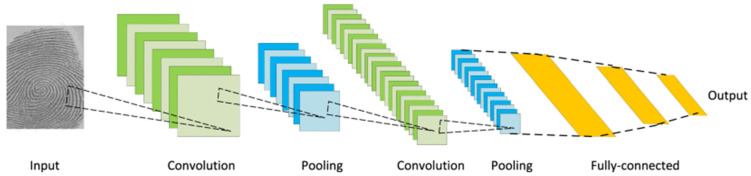
Generic architecture of CNNs [93].

**Figure 6 bioengineering-11-01034-f006:**
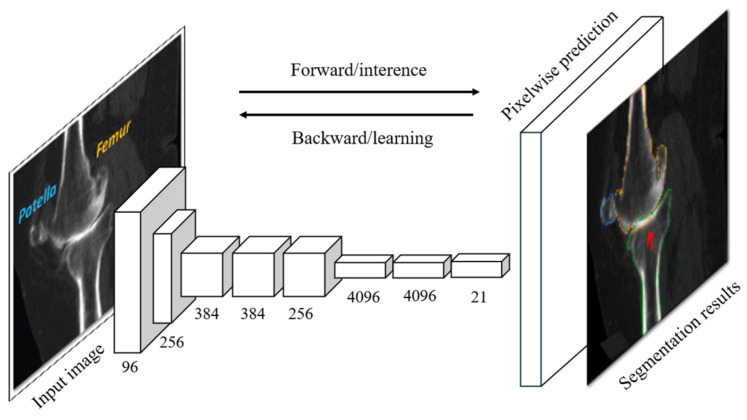
The architecture of FCNs demonstrating the forward inference process that generates pixel-wise predictions and the backward learning process for updating weights during training.

**Figure 7 bioengineering-11-01034-f007:**
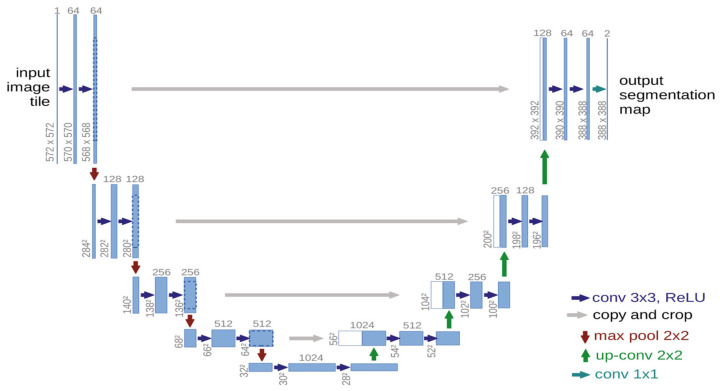
The structure of the U-Net [16].

**Figure 8 bioengineering-11-01034-f008:**
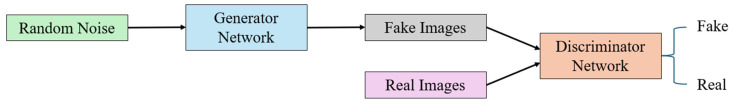
The structure of the GANs.

**Figure 9 bioengineering-11-01034-f009:**
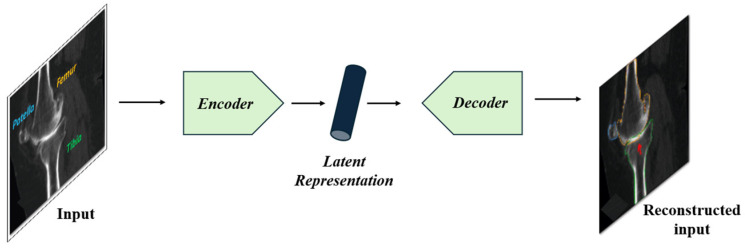
A generic architecture of AEs.

**Table 1 bioengineering-11-01034-t001:** Advantages and limitations of different edge detectors [44,45,46,47].

Operator	Advantages	Disadvantages
Roberts Cross	The computation is simple and effective in noise minimization scenarios.	Sensitive to noise and inaccurate positioning of precise edges.
Prewitt	Particularly sensitive to edge orientation	Susceptible to noise and less crisp edges.
Sobel	Similar to Prewitt but provides better noise suppression while also providing moderate edge smoothing.	Edges may be too blurred and unsuitable for accurate edge detection tasks.
Canny	Canny boasts a high accuracy in edge detection, efficiently suppressing noise, which contributes to its ability to precisely position edges.	Canny is computationally complex, and its performance is highly dependent on parameter selection.
Laplacian	Accurately identifies the center of edges and is sensitive to fine detail.	Highly sensitive to noise and does not provide information about edge orientation.
Laplacian of Gaussian	Preliminary Gaussian smoothing reduces the effect of noise and provides good edge localization.	Higher computational complexity and may miss some fine edges.

**Table 2 bioengineering-11-01034-t002:** A comprehensive summary of various graph-cutting methods, delineating each method’s cost function, optimization technique, foundational principles, and reference.

Method	Cost Function	Optimization Method	Bias	Reference
Minimal cut	$c(A,B|w\left(u,v\right))$	Gomory–Hu’s K-way maxflow algorithm	Short boundary	Wu and Leahy [75]
Ratio regions	$\frac{c\left(A,B|w\left(u,v\right)\right)}{\left|A\right|}$	Minimize the ratio between the cost of the bounding contour and the benefit of the enclosed region	Smooth boundary	Cox et al. [76]
Ncut	$\frac{c\left(A,B|w\left(u,v\right)\right)}{c(A,V|w\left(u,v\right))}+\frac{c\left(A,B|w\left(u,v\right)\right)}{c(B,V|w\left(u,v\right))}$	Measure both the total dissimilarity between the different groups as well as the total similarity within the groups	Similar weight partition	Shi and Malik [77]
Minimum mean weight	$\frac{\left|\int_{R}fdxdy\right|}{\int_{\partial R}gds}$	Globally Optimal Regions and Boundaries	Depends on f and g	Jermyn and Ishikawa [78]
Mean cut	$\frac{c\left(A,B|w\left(u,v\right)\right)}{c(A,B|1)}$	Global bipartition	None	Wang and Siskind [79]
Ratio cut	$\frac{c_{1}\left(A,B\right)}{c_{2}\left(A,B\right)}$	Global bipartition	None	Wang and Siskind [80]
Region-based Ncut	$N_{A}\cdot \frac{c\left(A,B\right)}{Totalw\left(A,V\right)}+N_{B}\cdot \frac{c\left(A,B\right)}{Totalw\left(B,V\right)}$	Consider the number of links on a Ncut basis	Similar weight partition	Cigla and Alatan [81]

**Table 3 bioengineering-11-01034-t003:** Performance summary of deep learning-based segmentation methods on public medical imaging datasets (TC: Tumor Core, WT: Whole Tumor, ET: Enhanced Tumor).

Anatomical Region	Deep Learning Method	Dataset	DSC	IOU	Recall	Accuracy	SE	SP
Brain tumour	CNN models [140]	BraTS dataset	/	0.633	/	0.927	/	/
	Patch-wise U-Net [141]	OASIS dataset	0.93	0.877	/	/	/	/
	BU-Net [138]	BraTS 2017 and 2018 datasets	TC: 0.901 WT: 0.837 ET: 0.788	/	/	/	/	/
	ACU-Net [142]	BraTS 2015, 2018, and 2019 datasets	WT: 0.9273 ET: 0.8429 TC: 0.9580	/	0.9415 0.8119 0.7856	0.9652 0.9054 0.9248	/	/
	AGResU-Net [143]	BraTS 2017–2019 datasets	TC: 0.787 WT: 0.876 ET: 0.745	/	/	/	/	/
	Inception U-Det [144]	BraTS 2020, 2018, and 2017 datasets	0.868	0.777	/	0.988	/	/
	ME-Net [145]	BraTS 2020 dataset	TC: 0.74 WT: 0.88 ET: 0.70	/	/	/	0.742 0.905 0.724	0.999 0.999 0.999
	AFP-Net [146]	BraTS 2013 dataset	TC: 0.73 WT: 0.85 ET: 0.67	/	/	/	/	/
Breast	SK U-Net CNNs [147]	UDIAT, OASBUD, and BUSI datasets	0.701	/	/	0.965	/	/
	CNNs [148]	DSprivate42 and DSpublic88 datasets	0.9578	/	/	0.9897	0.9549	0.8549
	Connected U-Nets [149]	CBIS-DDSM, INbreast datasets	0.9109	0.839	/	/	/	/
	Expanded U-Net [150]	Breast ultrasound image	0.905	0.827	/	/	/	/
	MF U-Net [151]	BUSIS datasets	0.9535	0.9112	0.9421	/	/	/
	U-Net [152]	Kaggle dataset	/	/	/	0.942	/	/
	GG-Net [153]	DAF’s dataset	0.954	0.912	0.957	0.951	/	/
	U-Net [154]	TCGA-BRCA dataset	0.680	/	/	/	/	/
	SMU-Net [155]	BUS datasets	0.8078	0.7012	/	0.7474	0.8933	/
	ASSGAN [156]	DBUI, OASBUI, SPDBUI and SDBUI	0.8319	0.7123	/	0.9589	/	/
Liver tumour	Modified SegNet [157]	3D-IRACDb dataset	/	0.937	/	0.988	/	/
	CEDCNN [158]	3DIRCADb dataset	0.9522	/	/	/	/	/
	ResU-Net [159]	3D-IRCADb01 dataset	0.97	/	/	/	/	0.9618
	Dilated convolutions [160]	3DIRCADb dataset	0.9638	/	/	/	/	/
	DRAUNet [139]	LiTS, 3DIRCADb, and Sliver07	0.972	/	/	/	/	/
	Mask-RCNN [161]	CT images	/	/	/	0.878	/	/
	CEDRNN [162]	3DIRCADb dataset	0.952	0.91	/	0.957	/	/
	UNet++ network [163]	MRI images	0.915	/	/	/	/	/
Lung	Attention U-Net [164]	JSRT dataset	0.975	/	/	/	/	/
	Mask R-CNN [165]	CT scans	0.97	/	/	0.9768	0.8772	0.8670
	Residual U-Net [166]	LUNA16 VESSEL12, and HUG-ILD	0.9898	0.9798	0.9898	0.9898	/	/
	GAN-based model [167]	CXR datasets	0.9740	0.943	/	/	/	/
	3D-Res2Unet [168]	LUNA16 public dataset	0.953	/	0.991	/	/	/
	DC-U-Net model [169]	CT images	0.9743	0.9627	0.9699	0.9731	/	
	ResBCDU-Net [170]	LIDC-IDRI database	0.9731	/	0.9701	0.9758	/	/
	LGAN [171]	LIDC-IDRI dataset	0.985	0.978	/	/	/	/
	EfficientNet-B4 [172]	Benchmark lung segmentation	0.978	0.957	/	0.989	0.9795	0.989
Prostate	2D U-Net model [173]	Public dataset PROMISE12	0.89	/	/	/	/	/
	Deeply supervised U-Net [174]	Promise12 public dataset	0.9067	/	/	/	/	/
	3D–2D Hybrid UNet [175]	Prostate mpMRI	0.898	/	/	/	/	/
	3D AlexNet [176]	MRI images	0.9768	/	/	/	0.896	0.902
	MetricUNet [177]	PROMISE 2012 dataset	0.9	/	/	/	0.89	/
	HF-Unet [178]	CT images	0.88	/	/	/	0.88	/
	3D PBV-Net [179]	PROMISE 12 and TPHOH	0.9613	/	/	0.9797	/	/
	SegDGAN [180]	PROMISE12 public dataset	0.8869	/	/	/	/	/
Retinal Vessel	RCED-Net [181]	DRIVE, CHASE_DB1, and STARE	/	/	/	0.9333	0.8419	0.9801
	M-GAN [103]	DRIVE, STARE, HRF, and CHASE-DB1datasets	/	0.7198	0.8324	0.9761	0.8324	0.9938
	CNN-based method [182]	DRIVE and STARE datasets	/	/	/	0.9716	0.8096	0.9841
	SA-Unet [183]	DRIVE and CHASE_DB1 datasets	/	/	/	0.9724	0.8573	0.9835
	Lightweight Attention CNN [184]	DRIVE, STARE, AND CHASE_DB1	/	/	/	0.9627	0.8050	0.9817
	Pyramid U-Net [185]	DRIVE and CHASE-DB1 datasets	/	/	/	0.9615	0.8213	0.9807
	CRAUNet [186]	DRIVE and CHASE_DB1 datasets	/	/	/	0.9659	0.8259	/
	ResDO-U-Net [187]	DRIVE, STARE, and CHASE_DB1	/	/	/	0.9623	0.8015	0.9807
	Wave-Net [188]	DRIVE set, STARE set, and CHASE	/	/	/	0.9592	0.8046	0.9798
Skin lesion	UNet-SCDC [189]	ISIC Skin Lesion Challenge datasets	0.885	0.824	/	0.929	0.953	0.911
	IFCN [190]	IEEE ISBI 2017 Challenge and PH2	0.9302	0.871	/	0.9692	0.9688	0.9531
	CNN [191]	ISBI 2016 and 2017 datasets	0.862	0.783	/	0.938	0.870	0.964
	Res-Unet [192]	ISIC 2017 dataset	0.924	0.854	/	/	/	/
	CNN [193]	HAM10000 dermoscopic image	/	/	0.8480	0.961	/	/
	MSAU-Net [194]	ISIC 2017, ISIC 2018, and PH2	0.9377	0.9617	/	0.9617	0.943	0.9698
	U-Net [195]	PH2 dataset	/	0.976	/	0.977	/	/
	Ms RED [196]	ISIC 2016, 2017, 2018, and PH2	0.8999	0.8345	0.9049	0.9619	/	/

**Table 4 bioengineering-11-01034-t004:** Comparative analysis of traditional and deep learning techniques in medical image segmentation.

Technique	Advantages	Disadvantages	Suitability for Specific Medical Applications
Thresholding	Simple, fast, computationally efficient [204]	Limited by stray colors and intensity variations, it does not work well for complex images	Works well in high-contrast regions (e.g., bones, lungs) [205,206]
Edge-based Segmentation	The effective detection boundary is clear and highly interpretable	Sensitive to noise and struggles with weak or fuzzy edges [207]	Excellent for tumor boundary detection in MRI and CT images with sharp edges [208]
Region-based Segmentation	Effectively detects homogeneous regions and excels at capturing local continuity	Dependence on seed point selection makes it difficult to deal with heterogeneous regions [209]	Available for organ and tissue segmentation with minimal intensity variation, such as the liver in CT [210]
Clustering-based Segmentation	Can group similar pixels without prior knowledge [211]	Sensitive to noise; may require post-processing	Tumor segmentation where intensity contrasts with surroundings [212]
Graph-based Segmentation	Handles global image context well, flexible handling of complex structures [213]	Computationally intensive and sensitive to the quality of graph construction [70]	Complex tissue structures (e.g., brain MRI) [214]
CNNs	Learns spatial hierarchies automatically, scalable [215]	Requires large labeled datasets and high computational power [216]	Tumor segmentation, lesion detection (e.g., brain, liver) [217]
FCNs	Pixel-level segmentation, preserves spatial resolution [218]	Sensitive to dataset size and variability [219]	Accurate pixel-wise segmentation (e.g., retinal vessels) [220]
U-Net	Strong performance with small datasets, effective in medical images [221]	Memory-intensive, requires much tuning for optimal performance [222]	Medical imaging, especially in biomedical fields (e.g., prostate, skin lesions) [195,223]
GANs	Generates high-quality, refined segmentations [224]	Challenging to train, requires large datasets, prone to instability in training [225]	Generating realistic medical segmentations (e.g., skin lesions) [189]
RNNs	Captures temporal dependencies, can handle sequential imaging data [226]	Difficult to train, prone to vanishing gradient problems, computationally heavy [227]	Volumetric image segmentation (e.g., heart, liver) [228,229]
AEs	Excellent for feature learning and dimensionality reduction [230]	Limited for highly complex segmentation tasks, lower accuracy than CNN-based methods	Feature extraction, anomaly detection (e.g., MRI, CT scans) [231]

## Data Availability

Data are fully available upon reasonable request to the corresponding author.

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
