# Peer review of "Advances in Medical Image Segmentation: A Comprehensive Review of Traditional, Deep Learning and Hybrid Approaches"

_bioengineering, 2024, doi:10.3390/bioengineering11101034_

Round 1

Reviewer 1 Report

Comments and Suggestions for Authors

References 1,8, 11, 12, 50, 55, 84, 92, 102, 114, 122, 123, 126, 141, 144, 151, 161, 162, 163, 164, 165, 166 , 176, 179, 183, 186, 187, 193, 197, 199, needs to be revised – provide names of all the authors rather than et al

Reference 21, name of the journal is missing, rather name of authors are repeated

Line 41 (introduction), citation can be provided

In section 3.2, final paragraph 572-578 can be revised, these are very generic statements and overfitting can be managed using various methods which is evident from studies reviewed in 3.3

Discussion/section 5 can be more elaborated where authors need to provide their own views and future directions.

Reviewer 2 Report

Comments and Suggestions for Authors

This article is a comprehensive review of medical image segmentation techniques, focusing on the applications and advances in the field of traditional methods, deep learning methods, and hybrid methods of both. The first is an overview of five classes of traditional image segmentation techniques, noting that these methods are computationally efficient and interpretable, but often face challenges when dealing with complex noisy or variable medical images. The second is a detailed discussion of several deep learning architectures for specific applications in medical image segmentation, but the learning models require large amounts of labeled data and high computational resources. By analyzing examples of deep learning in different clinical applications, such as segmentation of brain tumors, breast, and liver tumors, the potential and effectiveness of deep learning techniques in real-world medical image analysis are demonstrated. Finally, the method of combining deep learning with traditional segmentation methods is pointed out to overcome their respective limitations. Through a comprehensive analysis of current research progress, the article provides valuable insights into future research directions in the field of medical image segmentation, which may guide future research and development and drive further technological advancement in the field.

The article also has some shortcomings.

1.The challenges posed by the generalization ability of deep learning models and the high demand for computing resources were mentioned, but specific strategies for addressing them were not provided in depth.

2.Insufficient depth in comparative analysis between different technologies may hinder a clear understanding of their respective advantages and disadvantages, as well as their suitability in specific application scenarios.

3.The performance evaluation of medical image segmentation relies on various indicators, yet the article fails to provide a unified and standardized assessment of all the methods mentioned.

4.In the final section, future research directions were mentioned, but these directions are relatively broad and lack specific and actionable recommendations. Future research should focus on improving the efficiency and interpretability of deep learning models and developing more optimized hybrid methods.

Based on the reasons above,  the manuscript should be improved before accepted for publication.
